# Thermodynamically controlled multiphase separation of heterogeneous liquid crystal colloids

Han Tao [1], Carlo Rigoni [2], Hailong Li[3], Antti Koistinen [1], Jaakko V. I. Timonen [2], Jiancheng Zhou[4], Eero Kontturi [1] ✉, Orlando J. Rojas [1,5] ✉ & Guang Chu [1,4] ✉

Phase separation is a universal physical transition process whereby a homogeneous mixture splits into two distinct compartments that are driven by the component activity, elasticity, or compositions. In the current work, we develop a series of heterogeneous colloidal suspensions that exhibit both liquid-liquid phase separation of semiflexible binary polymers and liquid crystal phase separation of rigid, rod-like nanocellulose particles. The phase behavior of the multicomponent mixture is controlled by the trade-off between thermodynamics and kinetics during the two transition processes, displaying cholesteric self-assembly of nanocellulose within or across the compartmented aqueous phases. Upon thermodynamic control, two-, three-, and four-phase coexistence behaviors with rich liquid crystal stackings are realized. Among which, each relevant multiphase separation kinetics shows fundamentally different paths governed by nucleation and growth of polymer droplets and nanocellulose tactoids. Furthermore, a coupled multiphase transition can be realized by tuning the composition and the equilibrium temperature, which results in thermotropic behavior of polymers within a lyotropic liquid crystal matrix. Finally, upon drying, the multicomponent mixture undergoes a hierarchical self-assembly of nanocellulose and polymers into stratified cholesteric films, exhibiting compartmentalized polymer distribution and anisotropic microporous structure.

Phase separation is a persistently relevant topic that essentially entails order appearing from disorder in a heterogeneous system[1]. It can be triggered by different stimuli, occurring in a diverse range of natural and synthetic transition processes, including purification, compartmentation, and matter exchange[2–4]. To minimize free energy, certain kind of colloids can demix into two coexisting phases to reach equilibrium[5]. Specifically, dispersions of anisotropic particles undergo

phase separation above the critical concentration[6–9], termed as liquid crystal phase separation (LCPS) which has an immense potential in modern materials science with a wide range of applications[10]. In another realm, liquid-liquid phase separation (LLPS) in homogeneous macromolecular solutions is a prominent phenomenon in biological systems as it plays a dominant role in realizing cellular functions and creating subtle structures in living organisms[11]. Furthermore, both of

[1]Department of Bioproducts and Biosystems, Aalto University School of Chemical Engineering, Vuorimiehentie 1, 02510 Espoo, Finland. [2]Department of Applied Physics, Aalto University School of Science, Puumiehenkuja 2, 02150 Espoo, Finland. [3]State Key Laboratory of Fine Chemicals, School of Chemical Engineering, Dalian University of Technology, Dalian 116024, China. [4]School of Chemistry and Chemical Engineering, Southeast University, Nanjing 211189, China. [5]Bioproducts Institute, Department of Chemical & Biological Engineering, Department of Chemistry and Department of Wood Science, The University of British Columbia, 2360 East Mall, Vancouver, BC V6T 1Z3, Canada. ✉e-mail: eero.kontturi@aalto.fi; orlando.rojas@ubc.ca; chuguang88@gmail.com

these two transitions can be mutually connected: in eucaryotic cells, for example, the membranes are formed by LCPS of lipid molecules whereas the nucleolus is condensed from LLPS of proteins[12,13].

LCPS into upper isotropic and bottom ordered anisotropic phases is driven by the interplay between particle orientational entropy and excluded volume packing entropy, where the boundary conditions are governed by the aspect ratio of the particle (Fig. 1a)[14]. By contrast, the underlying driving force of LLPS is the trade-off between enthalpy and entropy toward energy minimization, based on the interactions between different components[15–17]. LLPS is controlled by a phase equilibrium which depends on, e.g., temperature, pH and/or composition (Fig. 1b)[18]. Once the phase separation occurs, the formed liquid droplet compartments exhibit varying physical properties and assume different colloidal states, such as gel, glass, or crystals[19–21]. Converging LCPS and LLPS has rarely been attempted in a synthetic system, although it would add a significant degree of tunability into LCPS if the physical triggers of LLPS, like temperature, were to be utilized in controlling their self-assembly behavior.

In this study, we report a thermodynamically controlled multiphase separation with cholesteric liquid crystal assembly in a four-component soft matter system. Specifically, the system consists of an aqueous mixture of cellulose nanocrystals (CNCs), poly (ethylene glycol) (PEG) and dextran. CNCs are rigid, polydispersed rod-like nanoparticles that are isolated from plant cell walls through acid hydrolysis[22–25]. Their colloidal stability and self-assembly ability to form left-handed cholesteric liquid crystal phases through the nucleation of tactoids have been widely explored[26–31]. Upon LCPS, the resulting

water-water interface exhibits ultralow interfacial tensions between the top isotropic and the bottom anisotropic phases[32,33]. PEG and dextran, on the other hand, are well-known for forming binary polymer mixtures which undergo a reversible LLPS transition process[34]. Intriguingly, this LLPS is temperature sensitive: it can be triggered by cooling when shifting the PEG-dextran composition from one-phase into the region that favors phase separation in the phase diagram[35,36]. In this way, a PEG-dextran solution with the composition at the intermediate region can separate into two phases at low temperatures and remix back into homogeneous state upon a temperature increase (Fig. 1c, d). The influence and colloidal behavior of adding guest additives into LCPS of CNC suspensions or LLPS of PEG-dextran solutions has been widely studied[37–42], also in the case of mixing the two systems with each other. Previously, we have established that the CNC nanoparticles can maintain their cholesteric order with the existence of PEG and dextran, while the resulting two CNC-polymers dispersions are immiscible when mixed, giving rise to water-in-water emulsions comprising hierarchical CNC assemblies[43,44]. There are, however, no attempts with any materials to create and investigate a heterogeneous colloidal system that features both the concentration dependent LCPS of nanoparticles and the temperature-sensitive LLPS of polymers.

By manipulating the CNC-PEG-dextran ratios, we reveal how the resulting mixtures exhibit two different equilibrium states with unusual phase coexistence behaviors and rich liquid crystal stackings by temperature adjustments. Owing to the interplay between suspended particles and polymer coils, the obtained LCPS-LLPS multiphase separation has displayed designed structural heterogeneity and

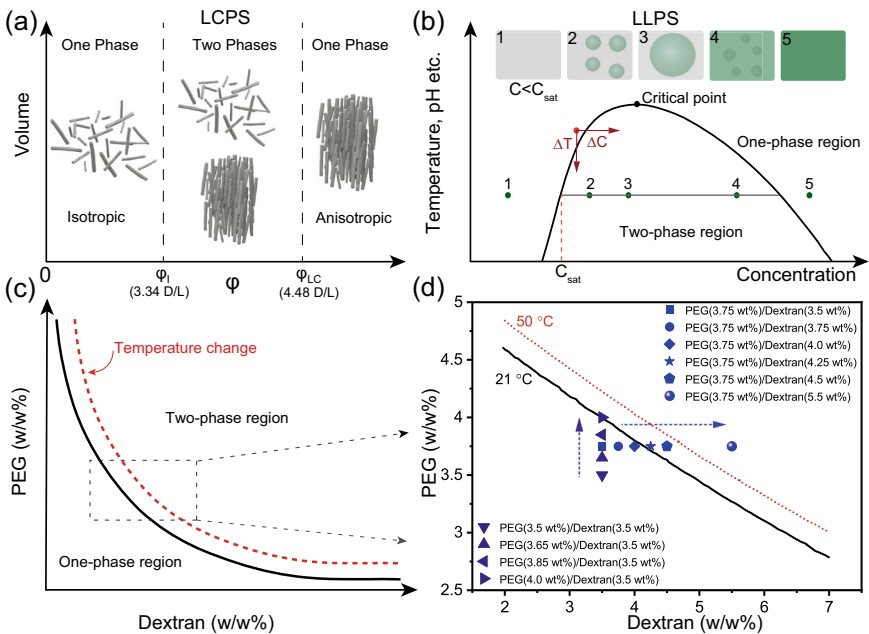

**Fig. 1 | Schematic illustration of the phase diagrams to describe LCPS of nanoparticles and LLPS of polymers. a** The volume-composition diagram illustrates two vertical lines at Onsager volume fractions $\varphi_I$ and $\varphi_{LC}$, corresponding to anisotropic colloidal particles with either one-phase or two-phase systems. **b** A phase diagram describes the phase behavior of macromolecular solutions as a function of concentration, influenced by various modulatory factors such as temperature and pH etc. The coexistence line (black curve) separates the one-phase region from the two-phase region on the phase diagram. At the critical point, the composition of the two liquid phases becomes identical, and the density difference between the phases approaches zero. Phase separation starting from a homogeneous state (red dot) can be induced by increasing the concentration ($\Delta C > 0$) or decreasing the temperature ($\Delta T < 0$). At concentrations below saturation concentration ($C_{sat}$), the system is in the one-phase region (1). Within the two-phase region, a polymer solution forms dense droplets with enriched solutes and a

depleted dilute phase (2). As concentration increases, droplets grow (3) until the dense phase volume surpasses the diluted, causing diluted droplets to form within the dense phase (4). Past this inversion, increasing concentration shrinks the diluted droplets until only a dense one-phase solution remains (5). **c** Phase diagram of PEG-dextran LLPS system: the binodal curve separates the phase diagram into a one-phase region that below the curve and a two-phase region that above the curve. The black curve is the coexistence line at the given temperature, whereas the red dashed curve represents a new coexistence line at an altered temperature. When the composition of PEG-dextran solution is located at the intermediate region between the two coexistence lines, it will stay in two-phase state at a given temperature and remix into one phase with temperature variation. **d** Experimental phase diagram of binary polymer solutions of PEG (8 kDa) and dextran (450–600 kDa) at 21 and 50 °C. The line data of critical PEG-dextran composition at different temperatures are obtained from a previous report[35].

ordered colloids assembly across the interface. Furthermore, we were able to distinguish different kinetic paths for LLPS and LCPS during the multiphase separation by disentangling the transition process into LLPS-induced polymer compartmentation and LCPS-driven chiral self-assembly of CNCs. When the mixtures were dried into solid films, the system underwent multiphase separation to preserve the cholesteric CNC assembly, leading to PEG-rich and dextran-rich polymer domains with varying helical pitch and hierarchical porous structure. The fundamental understanding provided by our results opens new directions for the design of novel hybrid soft matter with combined lyotropic and thermotropic responsivity.

## Results

In aqueous suspension, the CNC nanoparticles (195 ± 54 nm in length and 9 ± 2 nm in width, Supplementary Fig. 1) alone have purely repulsive interactions due to the strong anionic surface charges (zeta potential of -50 mV). Above the critical concentration, CNCs exhibit macroscopic liquid crystalline behavior with a bottom, high-density cholesteric phase and an upper, low-density isotropic phase (Fig. 2a). Typically, the phase behavior of CNCs is insensitive to temperature since it is driven by entropy (Supplementary Fig. 2). However, adding semiflexible nonadsorbing polymers, that is, PEG or dextran (molecular weight of 8 kDa and 450–600 kDa, respectively)[38,45] into the cholesteric liquid crystal phase can induce attractive interactions among lateral CNC nanorods by depletion mechanism and make the phase behavior stimuli responsive. In such aqueous CNC-polymer mixtures, CNCs are co-assembled with polymer coils into cholesteric organization and creating a region that is depleted of polymers around each nanorods (Fig. 2b and Supplementary Fig. 3). Owing to the molecular differences between PEG and dextran, the microscopic texture of CNC-PEG and CNC-dextran liquid crystal phase are slightly different under depletion effect (Supplementary Fig. 4). This maybe because of dextran (composed by branched glucose units) share similar molecular structure with CNCs (composed by linear chain glucose units), leading to more favorable steric and intermolecular interactions. Compared to the neat aqueous CNC suspension, the resulting overlapped depletion zones of neighboring nanoparticles in the CNC-polymer mixture drive the resuming cholesteric phase separation with reduced helical pitch (Supplementary Fig. 5 and Supplementary Table 1). In addition to concentration, the strength and scope of depletion interaction can be modulated by altering the size of the non-absorbing polymer[46–49], thereby tuning the hydrodynamic size of PEG and dextran through temperature adjustment enables the controlling of the phase behavior of the CNC-polymer mixture. Owing to the flexibility and expanded conformation of the polymer chains, when the cholesteric CNC-polymer mixtures are heated, both the anisotropic volume fraction and the helical pitch of the resulting cholesteric phase slightly decreases, which can be attributed to the changes of attractive depletion interactions between CNCs (Fig. 2c, Supplementary Fig. 6 and Supplementary Table 1). Therefore, we concluded that the LCPS of CNCs could be triggered by increasing concentration and further modified by adding semiflexible nonadsorbing polymers, giving rise to a concentration-driven, temperature-sensitive phase behavior.

In the experiments realizing multiphase separation, we used a four-component aqueous mixture comprising CNCs, PEG, and dextran at varying concentrations. We first examined the influence of LLPS of polymers on the LCPS behavior of CNCs by using a fixed PEG-dextran ratio of 3.75–4.25 wt%, while the CNC concentration was tuned from 0 to 6 wt% (Fig. 2d). After equilibrium, we determined the relative composition of each phase by size-exclusion chromatography and turbidity measurements (Supplementary Figs. 7–9 and Supplementary Table 2), indicating that the upper PEG-rich and bottom dextran-rich subsystems possess unequal affinity to CNCs. Note that the applied polymer composition (3.75–4.25 wt%) was located at the intermediate

region between the two coexistence curves, meeting the temperature conditions necessary for phase transition (Fig. 1d). In this case, the aqueous binary polymer mixture formed a two-phase system at 21 °C and was converted to a single phase at 50 °C, indicating the heating induced LLPS transition. The driving force behind this phenomenon can be attributed to the entropic gain associated with the release of water molecules from the polymer chains upon heating. However, the four-component CNC-PEG-dextran aqueous mixtures ($x$–3.75–4.25 wt%, $x = 1$–3) demonstrated a two-phase stacking without liquid crystalline ordering (Supplementary Fig. 10), implying that the addition of CNC had shifted the PEG-dextran composition in favor of a two-phase over the one-phase state in the phase diagram. Above the critical CNC concentration (4 wt%), both LLPS of polymers and LCPS of CNCs occurred in the hybrid system and revealed multiphase separation. At 21 °C, the dispersed CNCs were mainly partitioned into the bottom dextran-rich phase in which the depletion forces promoted the enrichment of particles together with cholesteric self-assembly. Meanwhile, the upper PEG-rich phase was CNC-lean and isotropic. The aqueous mixture separated into a three-phase system by a further increase in temperature to 50 °C with the stacking sequence including an upper PEG-rich isotropic phase, a middle dextran-rich isotropic phase, and a bottom dextran-rich cholesteric phase (Supplementary Fig. 11). Interestingly, in the presence of highly concentrated CNCs (6 wt%), the resulting multiphase coexistence behavior was temperature-dependent with either three-phase or four-phase stackings. Similar phase transitions and stacking sequences were obtained by the CNC-PEG-dextran mixtures with increased dextran concentration ($x$–3.75–5.5 wt%, $x = 0$–6, Supplementary Fig. 12), which was only for the two-phase region. These results suggested a thermodynamically controlled multiphase separation with individual LLPS of polymers and LCPS of CNCs in the heterogeneous colloidal system.

To explore the interplay between LLPS and LCPS, we selectively restrained the LLPS process by choosing a miscible PEG-dextran ratio of 3.75–3.5 wt%, and gradually increased the CNC concentration to realize the LCPS process (Fig. 2e). In such samples, the four-component aqueous mixtures remained homogeneous without phase separation at low CNC concentrations at both 21 °C or 50 °C. At intermediate CNC concentration (4 wt%) under room temperature, the aqueous mixture separated into two bulk phases, leading to an upper clear PEG-rich isotropic phase and a bottom, cloudy dextran-rich cholesteric phase, whereas phase separation was inhibited when the temperature increased to 50 °C (Supplementary Fig. 13). Higher CNC concentrations increased the volume fraction of the cholesteric phase and shifted from two-phase to three-phase stacking at room temperature. However, such system displayed an inversed phase separation phenomenon with the CNC partitioned into an upper low-density PEG-rich phase on top of a high-density dextran-rich isotropic phase (Supplementary Table 2). In parallel, further tuning of PEG-dextran polymer compositions (for example, dextran fixed at 3.5 wt% with PEG ranged from 3.5 wt% to 4.0 wt%) maintained LCPS with an increased CNC content whereas the polymer LLPS was suppressed (Fig. 1d and Supplementary Fig. 14). These results demonstrate a temperature-mediated multiphase separation, in which the lyotropic-induced LCPS of CNCs can be triggered by thermally controlled LLPS of polymers.

To quantify the interplay between LLPS and LCPS, we fixed the CNC-PEG concentration (6–3.75 wt%) and tuned the dextran concentration varied across two critical polymer compositions at different temperatures. The obtained mixtures exhibited three-phase stacking at 21 °C and two- or four-phase stacking at 50 °C, depending on the dextran concentrations (Fig. 2f, inset). For all the phase stackings investigated, the total volume fractions of the anisotropic phases slightly increased and did not coarsen with time (Fig. 2f and Supplementary Fig. 15), suggesting that they were equilibrated colloidal assemblies. By comparison, the temperature altered the volume fraction of the anisotropic phase, which could be ascribed to the size-

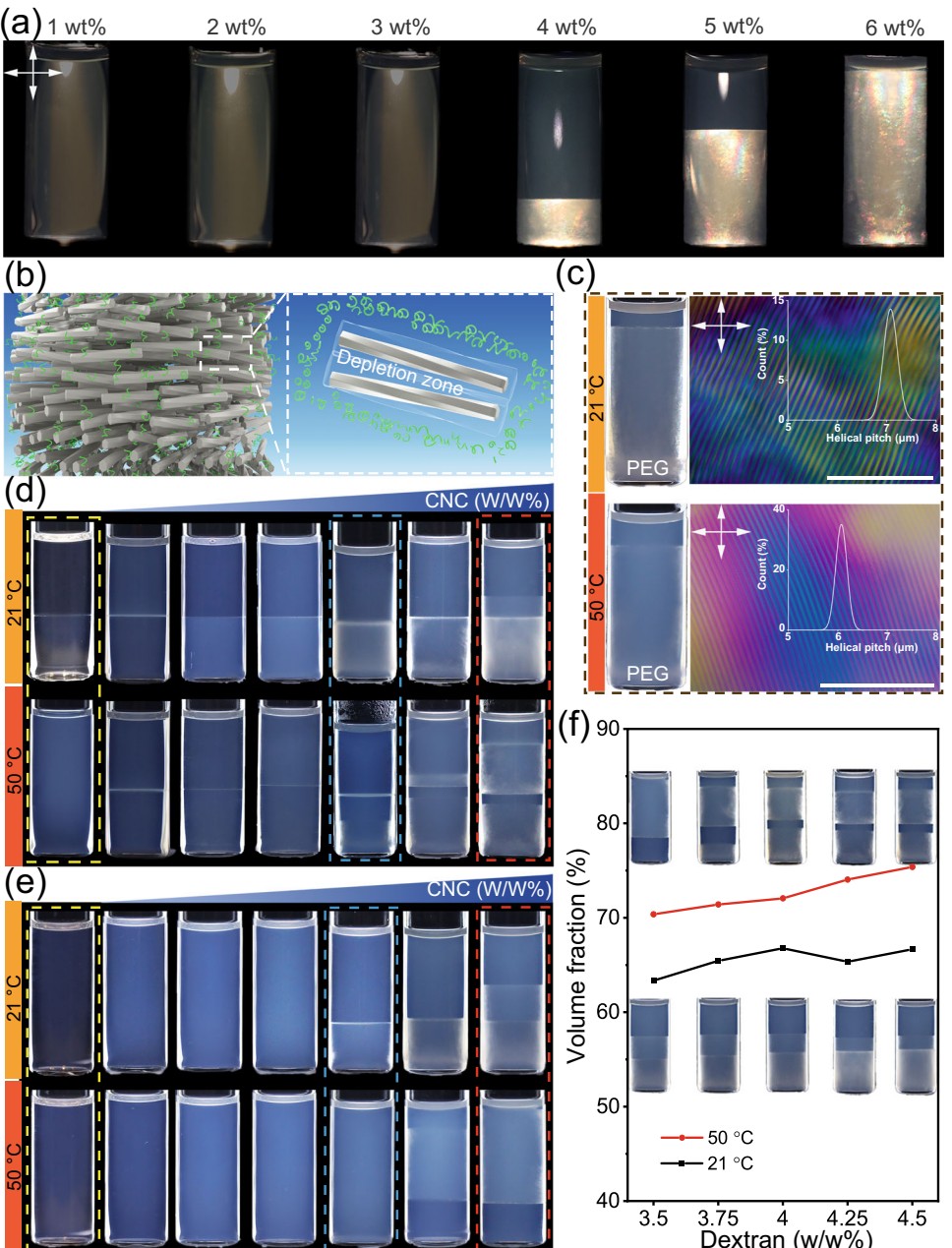

**Fig. 2 | The phase behavior of CNC-PEG-dextran heterogeneous colloids.**
**a** Photograph of aqueous CNC suspensions to highlight the concentration depen-dent LCPS (samples were placed between crossed polarizers). **b** Illustration of cholesteric self-assembly of CNC with nonadsorbing polymers through depletion effect. **c** CNC-PEG suspension with temperature-sensitive phase behavior and cholesteric assembly (scale bars: 50 μm). The inset is the corresponding measured helical pitch. **d**, **e** The evolution of multiphase separation in four-component CNC-

PEG-dextran aqueous mixture with an increased (stepwise 1 wt%) CNC content from 0 (yellow box), 4 (blue box, critical concentration for LCPS of CNC) to 6 wt% (red box) at different temperature. The composition of PEG-dextran is fixed at 3.75–4.25 wt% (**d**) and 3.75–3.5 wt% (**e**), respectively. **f** Total volume fraction of the anisotropic phase as a function of dextran concentration which equilibrated under either 21 or 50 °C.

dependent polymer depletion effect[50]. Besides, the resulting phase stacking structures were stable without further transformation when the temperature was tuned (Supplementary Fig. 16), implying the existence of an energy barrier between the different equilibrium states.

To better understand the multiphase separation, we clarify the interplay between LLPS and LCPS by mapping the three-dimensional phase diagrams for the four-component CNC-PEG-dextran aqueous mixtures at 21 and 50 °C, respectively (Fig. 3). When the CNC con-centration is below 3.25 wt% and equilibrate under 21 °C, the phase diagram of CNC-PEG-dextran aqueous mixture can be divided into

one-phase region and two-phase region by the coexistence plane (gray dots). The obtained mixtures remain homogeneous with the compo-sitions located at the left side of the CNC-PEG-dextran coexistence plane and phase separated due to the contribution of LLPS process at a composition on the right side of the coexistence plane (Fig. 3a). The phase separation behavior becomes more pronounced by increasing the CNC concentration, which is ascribed to the occurrence of LCPS. Given the multiphase separation process, the obtained CNC-PEG-dextran aqueous mixture display either a two-phase or three-phase behavior. It should be noted that the origin of the two-phase behavior can be realized by the contribution of LCPS or coupled LLPS-LCPS.

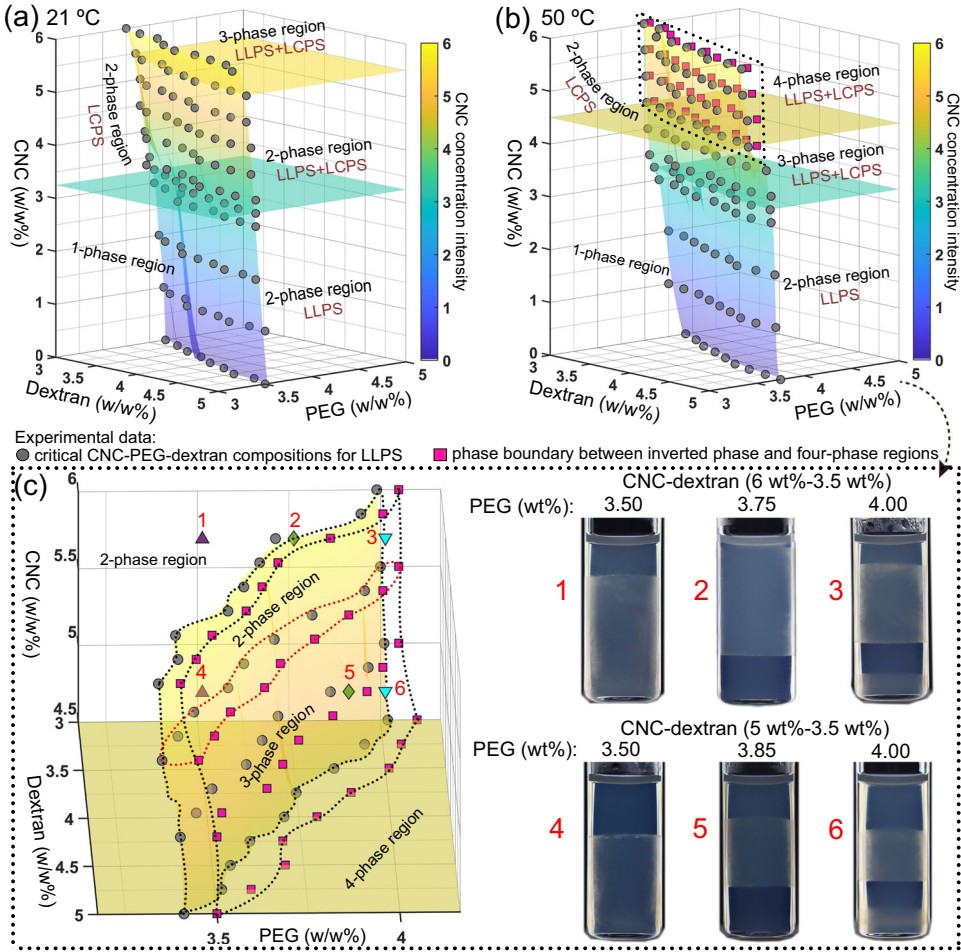

**Fig. 3 | Revealing the multiphase separation into different categories.** Three-dimensional phase diagram of the CNC-PEG-dextran aqueous mixture that high-light the interplay between LCPS of CNCs and LLPS of polymers at 21 °C (**a**) and 50 °C (**b**), respectively. Among which, the LLPS process is inhibited at the left side of the CNC-PEG-dextran coexistence plane (gray dots) whereas the right-side area is active for LLPS. With the contribution of LCPS, the resulting multiphase regions are modulated by the critical CNC concentrations (colored horizontal planes). **c** When the CNC concentration exceeds 4.5 wt%, the multiphase transition area can be defined as two-phase region (only LCPS, points 1 and 4), inverted phase region (where a cholesteric phase is on top of an isotropic phase, points 2 and 5), and a four-phase region (points 3 and 6) due to changes of the polymer composition.

Further increasing CNC concentration to 5.5 wt% leads to the emergence of an additional cholesteric phase within the PEG-rich region, thereby transforming the hybrid system from a two-phase to a three-phase state.

Upon heating, the interactions of CNC-polymers and PEG-dextran changed, leading to temperature-sensitive multiphase separation with two-, three- and four-phase coexistence behaviors of different physical origins (Fig. 3b). To explore the multiphase separation under 50 °C, the three-dimensional phase diagram can be partitioned by the CNC-PEG-dextran coexistence plane and the critical CNC concentration planes into several regions. With the addition of polymers, the LCPS of CNC is profoundly suppressed, namely, the CNC concentration threshold for LCPS shifts from 3.25 wt% (21 °C) to 4.50 wt% (50 °C) in the polymer miscible region. However, in the region where LLPS occurs, the critical concentration for LCPS of CNC remains at 3.25 wt%. Above this critical CNC concentration, the four-component CNC-PEG-dextran aqueous mixtures demonstrate three-phase behavior with LLPS and LCPS process, including an upper PEG-rich isotropic phase, a middle dextran-rich isotropic phase, and a bottom dextran-rich cholesteric phase. Intriguingly, when the CNC concentration reaches 4.50 wt%, the resulting mixtures exhibit more complex phase behavior with the emergence of varying inverted cholesteric-isotropic states. Figure 3c highlights the main critical coexistence plane (gray dots) and secondary phase plane (magenta square) with a transition from two-phase, three-phase to four-phase states, where the intermediate region between the two-phase boundaries displays an unique inverted phase behavior. Within this region, the volume fraction of upper cholesteric PEG-rich phase increases with the increasing of CNC concentration, and finally the multiphase system shifts from three-phase to two-phase stacking (from point 5 to 2). Further changes in the polymer concentration, below the main critical coexistence plane or beyond the secondary phase plane, leads to two-phase (point 1 and 4) and four-phase behaviors (point 3 and 6), respectively.

We validated the microstructure of each phase by performing polarized optical microscopy (POM) imaging. For the samples with high particle concentration, the resulting mixtures exhibited three-phase stacking at room temperature (Fig. 4a, b). The upper phase appeared dark while the bottom two phases displayed birefringence through crossed polarizers, confirming the isotropic (PEG-rich) | cho-lesteric (PEG-rich) | cholesteric (dextran-rich) stacking sequence. When viewed under higher magnifications, we observed two sharp interfaces at the isotropic-cholesteric and cholesteric-cholesteric domains in which the fingerprint texture in the latter region was continuously invaded from the middle cholesteric phase to the bottom phase, implying the chiral assembly of CNCs across the PEG-dextran phase boundary where the two phases are immiscible to each other (Fig. 4a, b, Supplementary Fig. 17). The measured helical pitch in the bulk phase of the cholesteric PEG-rich region was larger than that of the dextran-

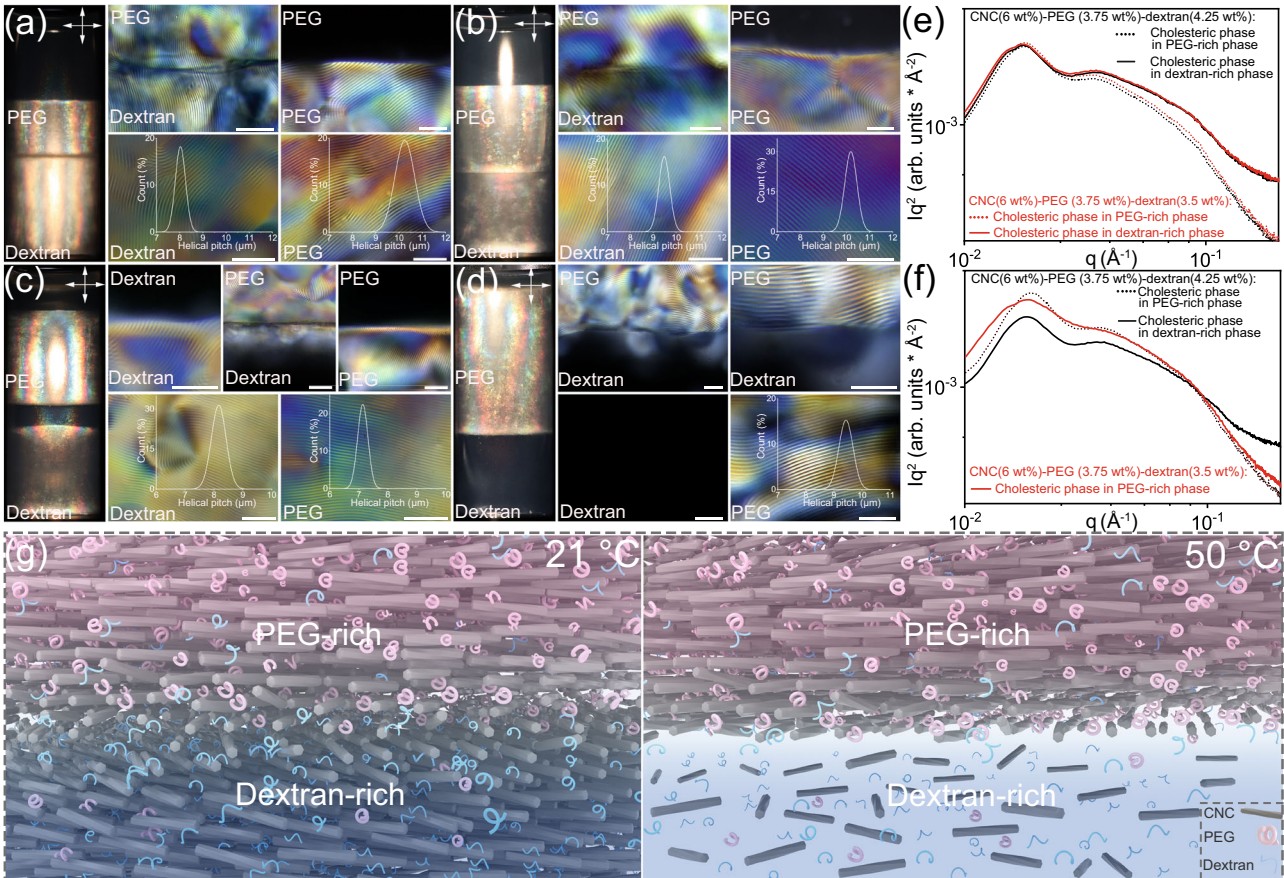

**Fig. 4 | From macro to micro-scales: POM imaging of the CNC-PEG-dextran four-component aqueous mixtures at equilibrium for different temperatures and focus on different interfaces (scale bars: 50 μm).** The initial composition of the mixtures is 6–3.75–4.25 wt% for (**a**, **c**) and 6–3.75–3.5 wt% for (**b**, **d**), where (**a**, **b**) are equilibrium under 21 °C and (**c**, **d**) are equilibrium under 50 °C, respectively. **e**, **f** The corresponding SAXS data of the four-component aqueous mixtures with different cholesteric phase stackings at 21 and 50 °C. **g** Schematic illustration of the PEG-dextran interface with cholesteric-cholesteric and cholesteric-isotropic stacking sequence across a gradient of polymer concentration.

rich region, implying a grading pitch along the polymer concentration variation which due to the differences of depletion interactions (Supplementary Table 2). When the samples were equilibrated at 50 °C, the obtained four-component aqueous mixtures transformed into a four-phase stacking with isotropic (PEG-rich) | cholesteric (PEG-rich) | isotropic (dextran-rich) | cholesteric (dextran-rich) assembly and two-phase stacking with cholesteric (PEG-rich) | isotropic (dextran-rich) assembly, respectively (Fig. 4c, d). Notably, the resulting helical pitch was temperature-dependent, which could be ascribed to the redistribution of polymers inside the cholesteric domains. Based on the difference between LLPS and LCPS, we distinguished two kinds of water–water interfaces in the hybrid system, namely, a demixed polymer-based PEG-dextran interface and a metastable liquid crystalline isotropic-cholesteric interface, respectively.

Along with the phase microstructure, we assessed the CNC-PEG-dextran self-assembly by small-angle X-ray scattering (SAXS). For three-phase stacking assembly, the obtained cholesteric phases displayed two peaks: a primary peak at low $q$ values fixed at 0.0178 Å$^{-1}$ and the peak at high $q$ values diminished in intensity from a dextran-rich to a PEG-rich phase (Fig. 4e). The average correlation distance of CNCs in the polymer-dispersed cholesteric phase was estimated to be 35.3 nm, i.e., slightly smaller than that of the neat CNC cholesteric phase (37.3 nm, Supplementary Fig. 18), implying the excluded volume effect of polymers on the cholesteric self-assembly of CNCs. With increasing temperature, the SAXS data of the corresponding polymer-dispersed cholesteric phase changed with the varying phase behavior (Fig. 4f). For four-phase stacking assembly, the primary peak position of PEG-rich cholesteric phase shifted to 0.0185 Å$^{-1}$ with a correlation distance of 34.0 nm, whereas the peak position of the dextran-rich counterpart remained constant (0.0178 Å$^{-1}$, 35.3 nm). For the two-phase assembly, the peak position and the correlation distance of the upper PEG-rich cholesteric phase was 0.0178 Å$^{-1}$ and 35.3 nm, respectively. Note that the twist angle between the neighboring CNCs in cholesteric phase can be determined by 360° × $d/p$, where $d$ and $p$ are the average correlation distance and helical pitch of the cholesteric phase, respectively. The calculated CNCs twist angles in different cholesteric regions ranged from 1.2° to 1.7°, suggesting slight changes of the helical twisting force with the coexistence of the polymers.

Based on the above, the schematic representation of the PEG-dextran interface with either cholesteric-cholesteric or cholesteric-isotropic assemblies, are shown in Fig. 4g. At the PEG-dextran interfacial zone, the relative polymer concentration gradually varies across distances of tens of nanometers[34,51], which is much larger than the distance between neighboring CNCs. The obtained two interfaces, namely, polymer-polymer and isotropic-cholesteric interface, could be coupled or separated by temperature changes, giving rise to ultralow interfacial tensions (<10 μN/m, Supplementary Table 3). For example, the interfacial tensions of the PEG-dextran interface with the cholesteric-cholesteric and the cholesteric-isotropic CNC phases were significantly reduced by the presence of the unequal partitioning of the negatively charged CNCs over the two phases.

To gain insights into the temperature-driven multiphase separation, we performed a detailed time-lapse imaging analysis that tracked the phase transition process. We first compared the dynamic formation of three-phase stacking (21 °C) and four-phase stacking (50 °C) with a CNC-PEG-dextran ratio of 6–3.75–4.25 (wt%). Figure 5a, b shows the time-resolved multiphase separation as well as the formation of interface at different temperatures (see Supplementary Movies 1, 2). For the three-phase stacking, the initial homogeneous mixture first became cloudy and sedimented into a bottom phase, which was ascribed to the contribution of LLPS with a dense lower dextran-rich phase and a dilute upper PEG-rich phase. Subsequently, the phase separation continued to reach equilibrium with the formation of a second phase on top of the PEG-dextran boundary that appeared less

cloudy while the upper phase remained clear, implying the isotropic-cholesteric LCPS transition. The resulting multiphase separation process was divided into two parts and took 20 h to reach equilibrium, longer than the individual LLPS of the PEG-dextran solution or the LCPS of the CNCs suspension (1.5 and 12 h, See Supplementary Movies 3, 4). On the other hand, when the sample was in equilibrium at 50 °C, the suspended CNCs partitioned into two compartments due to the LLPS of the polymers to arrive to a balance in chemical potentials between the two immiscible subphases. As the separation continued, the suspended CNCs in the PEG-rich and the dextran-rich phases then independently separated into an upper isotropic and bottom cholesteric phase separated by the PEG-dextran boundary, leading to a four-phase stacking assembly. For both multiphase separation systems,

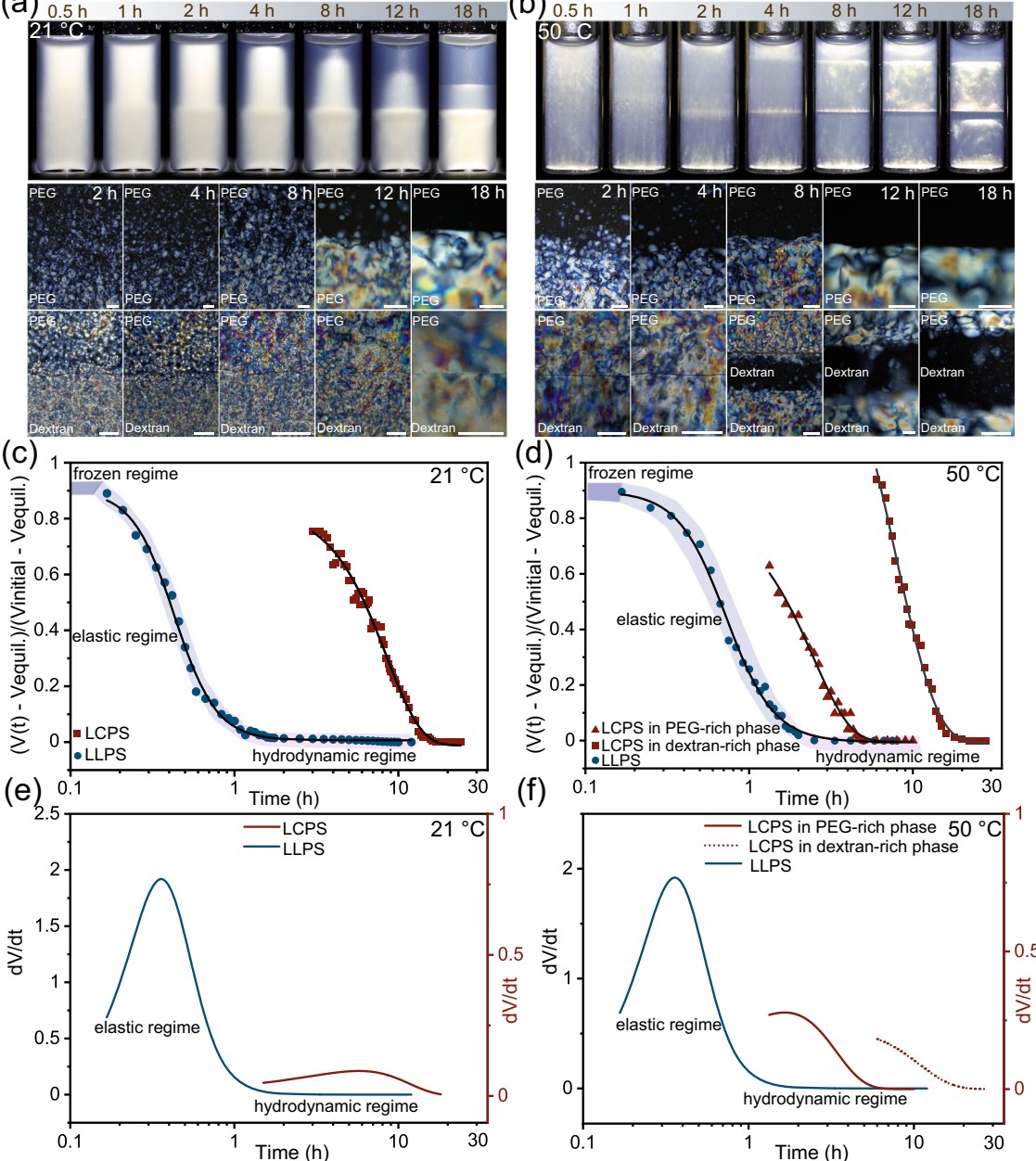

**Fig. 5 | Time evolution of the multiphase separation of a CNC-PEG-dextran four-component aqueous mixture (6–3.75–4.25 wt%) at different temperature.** Macroscopic pictures and the corresponding series of magnified POM images focus on the PEG-dextran phase boundary and isotropic-cholesteric liquid crystal interface during the multiphase separation at 21 °C (**a**) and 50 °C (**b**), respectively (scale

bars: 200 μm). Temperature-dependent multiphase separation kinetics measured under 21 °C (**c**) and 50 °C (**d**), displaying individual LCPS and LLPS processes. Phase separation velocity ($dV/dt$) of individual LLPS and LCPS transition calculated from kinetic data at 21 °C (**e**) and 50 °C (**f**).

LLPS led to an immiscible polymer boundary between the PEG-rich and the dextran-rich bulk phases before the terminating of LCPS of CNC. The microscopic LCPS was confirmed by the appearance of CNC cholesteric tactoids which accumulated into the isotropic-cholesteric interface under different temperature. Reaching equilibrium under 21 °C, the CNC tactoids accumulated into bulk cholesteric phase that diffused across the PEG-dextran boundary, resulting in a cholesteric-cholesteric interface at the polymer boundary and an isotropic-cholesteric interface in the upper PEG-rich phase. In contrast, the tactoids accumulated individually in the PEG-rich and the dextran-rich subphases and generated two isotropic-cholesteric interfaces in each compartment, whereas the polymer boundary was in a cholesteric-isotropic state at 50 °C.

The inherent difference between the three-phase and four-phase stackings was further investigated by analyzing their corresponding separation kinetics (Fig. 5c–f). The evolution of multiphase separation can be defined by $(V(t) - V_{equil})/(V_{initial} - V_{equil})$ as a function of time, where $V(t)$, $V_{initial}$ and $V_{equil}$ are the volume fraction of the corresponding phases at a given time $t$, at the initial state and at equilibrium state[52]. $V(t)$ is determined from the temporal evolution of the multiphase separation that disentangles into LLPS and LCPS steps. Therefore, we depict the real-time phase separation rate by $dV/dt$. As a complex fluid with binary polymer compositions, LLPS in PEG-dextran solution features asymmetric dynamics with both fast and slow components, displaying viscoelastic behavior due to the mismatch of different separation rates[53,54]. The measured separation kinetic for PEG-dextran solution showed a "fast" settling front of PEG-rich phase and a "slow" coalescing front of dextran-rich phase, typical of dynamic asymmetry in viscoelastic phase separation process (Supplementary Fig. 19). This dynamic asymmetry is induced by the differences in molecule size. For CNC-PEG-dextran mixture, adding CNCs can restrict the molecular motion and conformation of PEG and dextran, giving rise to a modified LLPS process. The resulting multiphase separation of LLPS step therefore can be divided into three parts: the initial frozen regime, the intermediate elastic regime and the final hydrodynamic regime (Fig. 5c, d)[53]. Once LLPS occurs, the system first becomes cloudy due to the nucleation of polymer-rich water droplets and shows no significant decreasing of volume fraction in the frozen stage. Then the volume fraction steeply decreases with time in the elastic regime. This phenomenon suggests the formation of network-like structure from the mechanical balance of elastic force (Supplementary Fig. 20), leading to a fast-coarsening mechanism with high separation rate (Fig. 5e, f). Due to the matter conservation, the changes of volume fraction in CNC-PEG-dextran system during LLPS stage indicate the selective transport of polymers through the immiscible phase boundary, namely, the nucleated dextran-rich droplets in the PEG-rich phase sediment downward and PEG-rich droplets in the dextran-rich phase float upward to across the boundary (Supplementary Figs. 21 and 22). In contrast to elastic period, the volume fraction of each phase remains stable in the hydrodynamic regime, and the network-like patterns is relaxed to spherical droplets due to the dominance of interfacial tension over elastic force (Supplementary Fig. 23). As a result, the LLPS process approaches equilibrium with the separation rate gradually decreases to zero (Fig. 5e, f).

As the multiphase separation continues, LLPS reaches equilibrium while LCPS is still evolving. The LCPS step showed one kinetic curve at room temperature and two independent kinetic curves for PEG-rich and dextran-rich compartments at 50 °C (Fig. 5c, d), implying the temperature-mediated LCPS that interacted with the PEG-dextran phase boundary. In principle, LCPS of CNCs into macroscale cholesteric bulk phase is the result of gravity-driven sedimentation and accumulation of microscale anisotropic tactoids through the continuous isotropic medium[30,31]. Therefore, the LCPS phase separation rate, i.e., the tactoid settling velocity, can be defined by $dV/dt = 2r^2 \Delta\rho g/9\eta$ (Stokes' Law), where $r$ is the diameter of the

tactoids; $\Delta\rho$ is the density difference between tactoids and the surrounding medium; $g$ is the gravitational constant, and $\eta$ is the viscosity of the surrounding medium[55]. The LCPS kinetics of pure CNC suspension showed a constant separation rate to approach equilibrium (Supplementary Fig. 19), indicating the Newtonian fluid behavior of the continuous isotropic medium. The addition of non-absorbing polymers of PEG and dextran generates a temperature-sensitive depletion attraction between CNC nanoparticles and significantly increase the viscosity of the continuous phase, thereby modulate the LCPS of CNCs. The LCPS kinetics of the four-component mixture underwent a steady separation rate, followed by an additional compaction stage with the rate gradually decreasing until equilibrium (Fig. 5e, f). Noted that the LCPS kinetics featured a fast separation rate in the upper, PEG-rich compartment and a slow rate in the bottom, dextran-rich compartment at 50 °C, presenting dynamic asymmetry between the components and characteristic for viscoelastic phase separation. We assumed that the average size of tactoids in each compartment was comparable, while the density difference $\Delta\rho$ was minimal between the isotropic medium and cholesteric tactoids. As a consequent, the resulting dynamic asymmetry inside the two polymer compartments during LCPS step is caused by their large viscosity difference between the upper and bottom continuous phase, namely, $\eta_{PEG} < \eta_{Dextran}$ (Supplementary Fig. 24).

According to classical thermodynamics, both LLPS of polymers and LCPS of nanoparticles can be described as activated processes, in which the critical nucleus begins with the formation of a new phase and requires an energy barrier[1]. For the current CNC-PEG-dextran system, the phase separation of PEG-dextran and CNCs itself can overcome their corresponding energy barriers through nucleation and growth, leading to a stable multiphase system that contains compartmented polymer-dispersed CNC liquid crystal colloids across the immiscible PEG-dextran phase boundary. The obtained LLPS and LCPS kinetics not only display strong dynamic asymmetry with viscoelastic phase behavior, but also widely differ in time across the varying separation stages, indicating an independent pathway for uncoupled LLPS and LCPS process.

With an understanding of the multiphase separation kinetics, we subsequently achieved a coupled LLPS-mediated LCPS process by manipulating the CNC concentration. Two representative temporal evolutions of the multiphase separation with different CNC-PEG-dextran ratios (4–3.75–3.5 and 6–3.75–3.5 (wt%)) are shown in Fig. 6. At low CNC concentration, the obtained colloidal suspension separated into two phases, with an upper isotropic and a bottom cholesteric phase at room temperature. However, it remained homogeneous when the temperature was increased to 50 °C, indicating temperature-driven phase separation that corresponded to LLPS and LCPS (Fig. 6a, Supplementary Fig. 13 and Supplementary Movie 5). The resulting kinetic data exhibited a similar trend as the kinetics of LCPS of CNCs (Fig. 6b), suggesting the dominant role of the LCPS during the multiphase separation process. Further insight into the colloidal self-assembly between the CNCs and the polymers was given by selectively dye-labeling the dextran with fluorescein isothiocyanate (FITC) molecules (FITC-dextran, Mw = 500 kDa). When viewed with differential interference contrast and fluorescent mode, a coupled LLPS-LCPS process was confirmed. We distinguished a typical nucleation of dextran molecules, rich in CNC cholesteric tactoids, from an isotropic background. The tactoids grew larger by merging and eventually sedimented downward into bulk phase. This is because of the higher density of dextran-rich tactoid than the PEG-rich dispersion medium, and the sedimentation rate is governed by Stokes' Law. Hence, the coupled LLPS-LCPS transition, driven by the sedimentation of tactoids, led to a dextran-rich cholesteric phase at the bottom and an upper PEG-rich, isotropic phase (Fig. 6c and Supplementary Fig. 25). By contrast, at 50 °C, the FITC-dextran molecules were uniformly distributed, without enriching in the interior of the tactoids

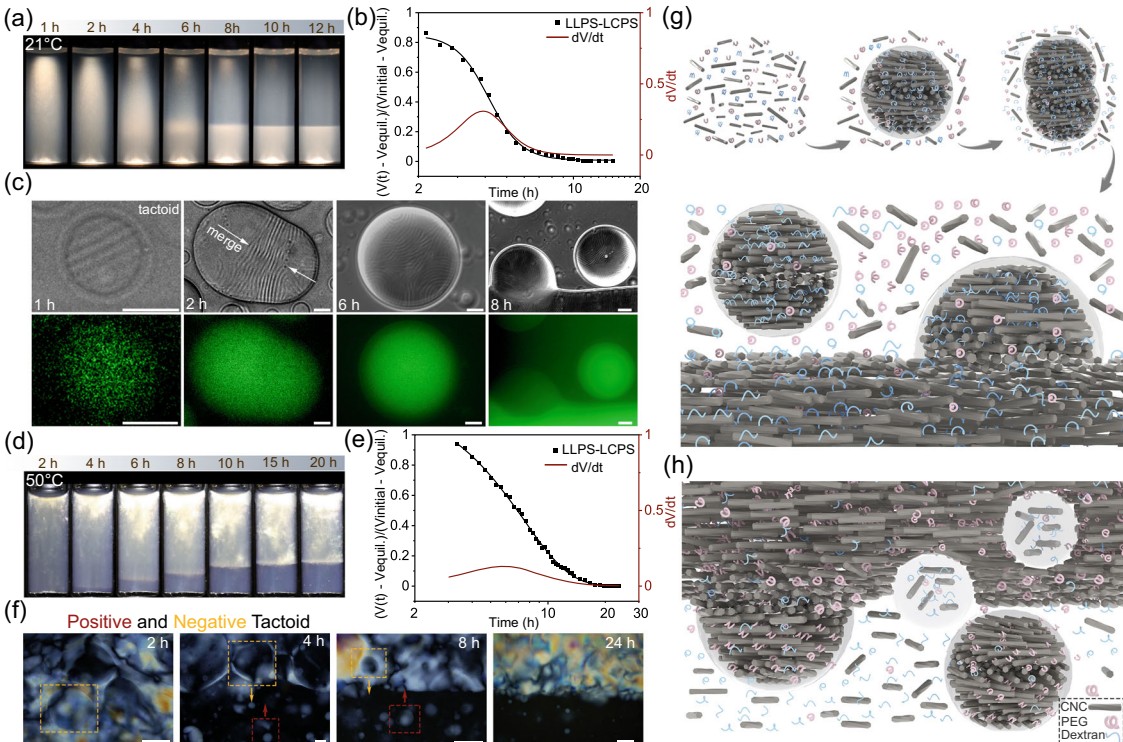

**Fig. 6 | The phase behavior of a CNC-PEG-dextran aqueous mixture with a coupled LLPS and LCPS process. a** Time evolution of the multiphase separation of CNC-PEG-dextran aqueous mixture (4−3.75−3.5 wt% at 21 °C). **b** Multiphase separation kinetic of the mixture that shows a coupled LLPS and LCPS process. **c** Differential interference contrast and fluorescence images of the multiphase separation through nucleation and grow process. The FITC-dextran molecules are enriched inside the cholesteric tactoids and merge into larger droplets, and eventually fuse into bulk dextran-rich cholesteric phase (scale bars: 50 μm). **d** Time evolution of the inversed multiphase separation with a composition of 6−3.75−3.5 wt% at 50 °C. **e** Inversed multiphase separation kinetic of the mixture. **f** Time-lapse POM images demonstrate the formation of negative (yellow rectangle) and positive tactoids (red rectangle) and their vertical movement during the multiphase separation, resulting in an inversed cholesteric-isotropic phase behavior (scale bars: 100 μm). **g** A schematic description of the coupled LLPS-LCPS process through nucleation and growth of polymers inside the tactoids. **h** Illustration of the inversed multiphase separation due to the accumulation of negative and positive tactoids.

(Supplementary Fig. 26), which could be attributed to the suppressed LLPS process, i.e., the formation of a miscible PEG-dextran polymer phase with discrete CNC cholesteric tactoids. Thus, we infer that both the LLPS and the LCPS processes occur simultaneously and share the same nucleation and growth pathway, in which the multiphase separation is triggered by LLPS but governed by LCPS transition.

We also tracked the inverse-phase separation process at 50 °C with the upper low-density PEG-rich cholesteric phase floating on the high-density dextran-rich isotropic phase at the bottom (Fig. 6d and Supplementary Movie 6). At the macroscale, the resulting mixture displayed a three-phase behavior, with two transition kinetic curves at room temperature, responsible for independent LLPS and LCPS process (Supplementary Fig. 27 and Supplementary Movie 7), whereas a coupled kinetic curve was obtained at 50 °C (Fig. 6e). In this respect, we proceeded to capture the structural evolution of the multiphase separation through a POM imaging sequence that highlighted the movement of tactoids at the liquid-liquid interface (Fig. 6f). Before separation, the mixture was homogeneous and birefringent, showing tortuous disclination lines and isotropic droplets surrounded by the fingerprint cholesteric domains. These isotropic droplets, termed as negative tactoids[56], grew larger and settling downward due to the higher density, merging into the bottom isotropic dextran-rich phase. As separation continued, the additional CNC and PEG molecules from the isotropic dextran-rich phase were nucleated into positive cholesteric tactoids, flowing-up to merge into the low-density PEG-rich cholesteric bulk phase. The vertical movements by tactoids that drove multiphase separation could be attributed to the density difference between the tactoid

droplets (including negative and positive) and their dispersion media which moved to opposite directions[27]. Once the system reached equilibrium, the compositions of each solute species satisfy the condition of equal chemical potential across the permeable polymer-polymer phase boundary[57]. Changing the temperature significantly modified the chemical potential in each solute, created multiphase separation and led to a coupled LLPS-LCPS process.

Based on above, Fig. 6g, h illustrates the two types of multiphase separation in the four-component mixture, enabled by the coupled LLPS-LCPS process. At low CNC concentration, entropic effects drive the nucleation of orientationally ordered cholesteric tactoids from a dense isotropic background. As a result, PEG and dextran molecules are unequally partitioning into the tactoids as they are maintaining the balance of the chemical potential. A bulk cholesteric liquid crystal phase forms through fusing and sedimentation of tactoid droplets, resulting in an upper isotropic PEG-rich phase and a bottom cholesteric dextran-rich phase. In other words, the LLPS of the polymers induce the LCPS of CNCs (Fig. 6g). However, when the CNC concentration is high enough, both isotropic and cholesteric tactoids are formed during the separation process. In this case, PEG and dextran molecules are selectively enriched inside the cholesteric and isotropic tactoids, respectively, and further merge into an upper cholesteric PEG-rich phase and a bottom isotropic dextran-rich phase (Fig. 6h). Overall, the resulting LLPS-LCPS multiphase separation follows the sedimentation kinetic of tactoid droplets, which indicates the dominant contribution of the LCPS process.

Owing to the LLPS of the polymers, the phase separated four-component mixture can be further used to generate hierarchical film

structures through water evaporation. Upon drying, the cholesteric order of CNC is preserved in solid films with the co-assembly of non-adsorbing PEG and dextran polymers. In comparison to the neat cholesteric CNC film or the binary CNC-polymer composite films, which display vivid structural color[58–61], the obtained CNC-PEG-dextran hybrid film appeared white under natural light and showed birefringent domains under POM imaging, indicative for anisotropic organization (Fig. 7a, b). Using FITC-dextran as the fluorescent tracer, we confirmed the stratified polymer distribution of PEG and dextran during the film preparation. Similar to the aqueous mixture, the solid composite was compartmentalized into PEG-rich and dextran-rich regions along the thickness of the film (Fig. 7c), which can be attributed to the polymer-polymer phase separation. The two CNC-polymer layers were packed together in the plane of the film with a thickness of 154 and 186 μm for PEG-rich and dextran-rich regions, respectively, consistent with their inclusion in the cholesteric phase within the CNC-PEG-dextran aqueous mixture.

The surface morphology and CNC cholesteric organization of the hybrid composite film was further observed by scanning electron microscopy (SEM). At low magnification, the morphology of the cross-section of the hybrid film exhibited a stratified structure with a dense PEG-rich domain and a microporous dextran-rich domain, respectively (Fig. 7d). Inspection of the PEG-rich domain at high magnification revealed a layered periodic structure of the cholesteric self-assembly of CNCs (Fig. 7e, f and Supplementary Fig. 28). Looking at the dextran-rich domain, we observed a hierarchical structure composed of a porous scaffold surrounded by a continuous network of buckled CNC layers (Fig. 7g). The cutting edge of the buckled layers displayed a well-aligned periodic arrangement of CNCs, characteristic of the cholesteric structure (Fig. 7h, i). The measured pitch of CNC cholesteric order in the PEG-rich and dextran-rich domains were 1075 and 850 nm, respectively, larger than the wavelength range of visible light. The

large voids (-5 μm in diameter) inside the network resulted from the formation and shrinkage of negative tactoids during evaporation, in which the isotropic PEG-rich droplets were surrounded by the dextran-rich CNC cholesteric phase (Supplementary Fig. 29). Therefore, we conclude that drying a four-component aqueous mixture of CNC-PEG-dextran can not only sustain the cholesteric organization of nanoparticles inside the polymer matrix, but also preserve the polymer-polymer phase separation in the solid film, leading to an anisotropic porous structure that allows maximizing the multiple scattering in the hybrid film with an opaque white appearance.

## Discussion

In summary, here we have bridged the gap between LLPS of polymers and LCPS of nanoparticles to achieve multiphase separation with thermodynamically controlled phase behavior. When the CNC-PEG-dextran aqueous mixtures are in equilibrium, the CNCs sediment into a cholesteric liquid crystal phase through nucleation and growth of tactoids, whereas the two polymers demonstrate tunable miscibility and create a permeable interface for CNC chiral self-assembly. By exploiting the influence of temperature, we show a responsive phase coexistence behavior with rich liquid crystal stackings that result from the interplay between LLPS and LCPS. In addition, systematic analysis of the evolution of multiphase separation indicates the varying paths of LLPS and LCPS processes which can be independent or coupled to each other. Upon drying the four-component mixture, a stratified cholesteric film with unequal polymer distribution was obtained, showing hierarchical porous structures and white appearance. Our findings outline a promising platform for designing novel heterogeneous complex colloidal systems whose functionalities can mimic and match those observed in biology.

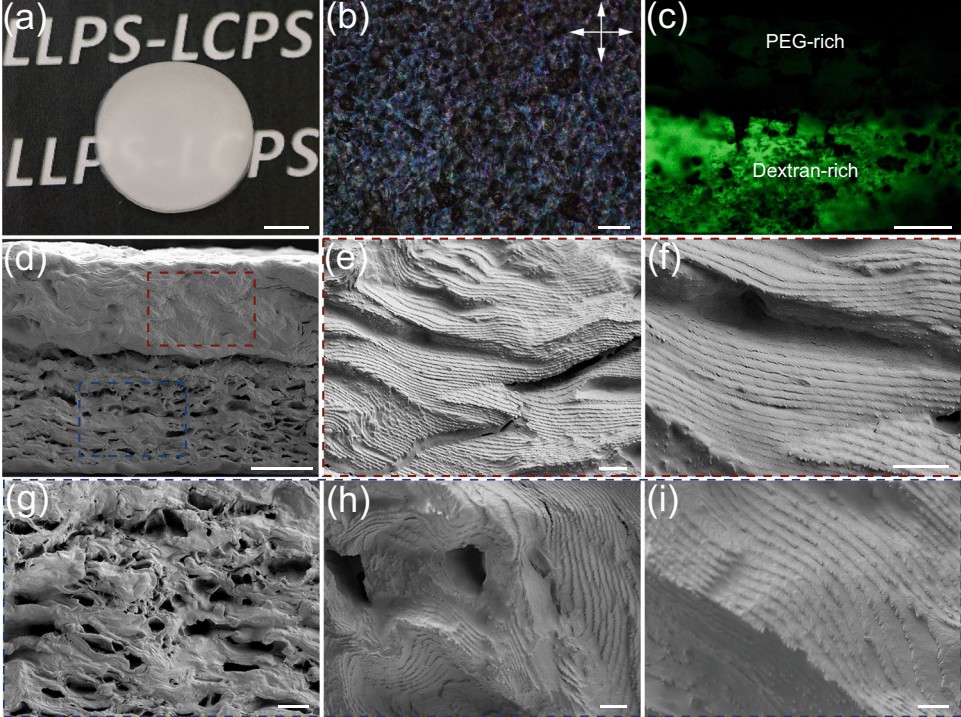

**Fig. 7 | Hierarchical structured cholesteric CNC-PEG-dextran composite film.**
**a** Photograph of a dried CNC-PEG-dextran hybrid composite with opaque white appearance. **b** POM image of the hybrid composite with distinct birefringent domains. Cross-section of the composite film with fluorescent (**c**) and SEM imaging (**d**), revealing highly dense morphology in PEG-rich region and microporous

morphology in dextran-rich region. **e**, **f** High magnified SEM images of the dense PEG-rich region that display periodic layered structure, indicative of cholesteric organization of CNC. **g–i** SEM images of the dextran-rich region, demonstrating deformed cholesteric layers around large pores. The scale bars are 1 cm (**a**), 200 μm (**b**), 100 μm (**c**), 100 μm (**d**), 4 μm (**e**), 2 μm (**f**), 20 μm (**g**), 2 μm (**h**), and 2 μm (**i**).

# Methods

## Materials and chemical reagents

Cellulose nanocrystals gel (CNC, 10.5 wt%) was purchased from the US Forest Products Laboratory at the University of Maine. Dextran (Mw = 450-600 kDa) and fluorescein isothiocyanate dextran (FITC-dextran, Mw = 500 kDa) were brought from Sigma-Aldrich (Helsinki, Finland), and polyethylene glycol (PEG, Mw = 8 kDa) with a purity ≥ 99% was obtained from Promega. All Chemicals were used as received without further purification. Milli-Q water used in all experiments was produced by a Millipore water system.

## Preparation of CNC liquid crystalline suspension

Two hundred milliliters of CNC suspension with a concentration of 6.5 wt% was prepared by diluting concentrated CNC gel with Milli-Q water. The as-obtained homogenous suspension was sonicated for 2 min in an ice bath to increase the particle dispersity. After equilibrating for 2 days (4 °C), the prepared CNC suspension separated into two phases with an upper isotropic and a bottom cholesteric organization. The upper isotropic phase was removed by pipette and leave the bottom cholesteric phase for subsequent experiments.

## Preparation of CNC-PEG-dextran heterogeneous liquid crystal colloids

In general, PEG and dextran powders with varying composition were dissolved into the CNC colloidal suspension with a certain amount of Milli-Q water to ensure the total mass of the mixture was maintained at 6 g (volume is around 5.3 mL). The obtained mixtures with desired compositions were magnetically stirred for 6 h at room temperature, and then transferred into the 8 ml vials for further usage. Multiphase separation of CNC-PEG-dextran four-component mixture was conducted under either 21 °C or 50 °C environments for two days to reach equilibrium, displaying temperature-sensitive phase behaviors.

## Preparation of CNC-PEG-dextran hybrid composite film

The freshly prepared CNC-PEG-dextran four-component aqueous mixture (6–3.75–3.5 wt%) was poured onto a Petri dish (60 mm in diameter) and sealed to rest for two days to reach the equilibrium. Then, this phase separated mixture was allowed to evaporate under ambient conditions to obtain the solid hybrid composite film.

## Morphology characterization of CNC

Transmission Electron Microscope (TEM) was used to characterize the morphology of CNCs. Measurement was conducted on a FEI Tecnai G2S-Twin (FEI, USA) with a field emission gun working at 200 kV. More than 100 individual crystallites in the TEM images were measured to gain the size distribution using ImageJ.

## Anisotropic volume fraction and time-dependent phase separation analysis

All photographs and time-lapse videos were shot by a single-lens reflex camera (Canon EOS 90d, Japan). Photographs of samples were captured as the phase separation completed in the suspensions at either 21 or 50 °C. Those obtained time-dependent images exhibit spreading of the supernatant/suspension interface that is used to differentiate the isotropic region and anisotropic region in the suspensions. Volume fraction of the anisotropic phase was determined by dividing its height by the total height of the dispersion using software ImageJ.

The time-dependent phase separation process was visualized by the time-lapse recording, and as-obtained videos were automatically transferred to the format of 25 FPS (1 s is equal to 12.5 min in real time) by a camera. To depicture the phase separation kinetics, time-lapse videos were transferred to image sequence stocks and those image sequences were imported into ImageJ. For each time step, the position of PEG-dextran phase boundary or isotropic-cholesteric liquid crystal interface was monitored in the software. The volume fraction of the corresponding phase at a given time was normalized by its height with respect to the height of the suspension.

## Polarized optical microscope imaging

For optical microscopic observation, 350 µL of the freshly mixed suspension was filled in the optical quartz cuvettes (1 mm path length, Hellma, Germany). Sealed cuvettes vertically rest to allow phase separation toward equilibrium. The anisotropy of the suspensions under equilibrium was verified by an Olympus BX53-P microscope (Olympus, Japan) in which each phase was imaged by a pair of polarizers in a perpendicular arrangement. Particularly, the phase boundary was captured by a homemade microscope with a 10× objective lens. In which, the cuvettes were positioned vertically between two crossed polarizers, and the incident light penetrated the suspension from right to left. Temperature control was conducted on Linkam PE120 heating stage (Linkam Scientific Instruments, UK). Evaluation of helical pitch of the cholesteric phase was conducted in ImagJ with measuring at least 100 points in the corresponding POM images.

## Fluorescence microscope imaging

To make the sample fluorescent, 0.01 wt% of FITC-dextran was mixed into the CNC-PEG-dextran three-component mixture (4–3.75–3.5 and 6–3.75–4.25 (wt%), respectively). The as-prepared dye-labeled mixture was filled in the glass capillary and sealed by a hot glue. The samples were vertically equilibrated to allow for multiphase separation under either 21 or 50 °C. A Zeiss Axio Vert A1 Inverted Microscope (Zeiss, Germany) with GFP setting was used to track the fluorescence signals.

## Small-angle X-ray scattering measurement

Small-angle X-rays scattering (SAXS) experiments were conducted in a transmission mode of Xeuss 3.0 on Xenocs CuKa x-ray instrument operated at 50 kV and 0.6 mV. Prior to measurements, the suspensions were filled into borosilicate capillaries with a diameter of 1.5 mm and a 10 µm wall thickness (Hilgenberg GmbH, Germany) for several days until phase separation completed. The sample detector distance was 500 mm, and exposure time was 1800 s.

## Hydrodynamic radii measurements

The hydrodynamic radii of CNC, PEG and dextran under either 21 or 50 °C was measured using Zetasizer Nano (ZS-90, Malvern Instruments, Worcestershire, UK).

## Scanning electron microscope imaging

Cross-section morphology of the CNC-PEG-dextran hybrid composite were characterized using a Zeiss Ultra Plus high-resolution scanning electron microscope (HR-SEM) at an accelerating voltage of 3 kV.

## Determine the composition of the multicomponent mixture

Due to the fact that the number of particles has a qualitative correlation with the turbidity, the relative CNC particles proportion in each phase can be estimated by its relative turbidity ratio. To measure the partitioning of CNC among the phases of the suspension, aliquots of each phase were carefully pipetted out and diluted to a desired concentration. Subsequently, the turbidity of suspension was measured by a UV–Vis spectrophotometer (UV-2550 Shimadzu) at 500 nm[62]. The turbidity of initial suspension was set as the reference value ($Turbidity_{initial}$) and $Turbidity_x$ was defined as the turbidity of the target phase. The normalized turbidity obtained by the equation of $Turbidity_x/Turbidity_{initial}$ can present the relative proportion of CNCs in separated phases.

The size-exclusion chromatography (SEC) experiments were conducted on each separated phase to determinate relative concentration of dextran and PEG. The experiments run in 0.1 M NaNO₃ with Agilent 1260 Infinity II Multi-Detector GPC/SEC System including refractive index from light-scattering (two measurement angles: 15°

and 90°) refractive index, and differential viscometer (VISC). Three water Ultrahydrogel columns (500 Å, 250 Å, and 120 Å) with a Ultrahydrogel guard column were mounted for separation with the flow rate of 0.5 mL/min. the separated phases were diluted 10 times with 0.1 M NaNO₃ eluted solution and 100 μl polymer solution was injected to the system. The detectors were calibrated using narrow dispersity pullulan standard with nominal $M_w$ of 110 kg/mol. Agilent OpenLAB CDS ChemStation Edition software was used for instrument control and Agilent GPC/SEC software for data collection and handling. Refractive index increment of 0.144 ml/g was used for and 0.134 ml/g for PEG.

The standard curves for PEG and dextran were established separately to calculate the relative quantities of PEG and dextran in each phase of the CNC-PEG-dextran mixture that equilibrated at different temperature. The RI peaks area ($A_{RI}$) were extracted from original figures by integrating individual RI peaks and plotted with the injected polymer concentrations. Linear relations ($C = k \times A_{RI}$) see in both PEG and dextran standard curves over a broad range of concentration from 0.5 to 10 mg/mL (Supplementary Fig. 7b). Therefore, the concentration of PEG and dextran in each individual phase can be calculated from the equation[63]:

$$C = n \times k \times A_{RI} \qquad (1)$$

In which, $n$ is the dilution factor, $k$ is 0.0491 for PEG and 0.04025 for dextran, respectively.

### Interfacial tension measurements

The surface tension of each phase was measured using a Theta Flex optical tensiometer (Biolin Scientific, Finland). A 12 μl of pendent drop squeezed by the device was analyzed at room temperature while a shield put on top of the sample stage to minimize the influences caused by the air flow from ventilation. The water-water interfacial tension between two adjacent phases can be estimated by the Good-Girifalco equation:[64]

$$\gamma_{12} = \gamma_1 + \gamma_2 - 2\varphi\sqrt{\gamma_1\gamma_2} \qquad (2)$$

Where $\gamma_1$, $\gamma_2$ represent the surface tension of the two adjacent phases, respectively, and $\varphi$ is the Good's interaction parameter. Because the interfacial tension between the PEG-dextran and isotropic-cholesteric interface is ultralow, here we used $\varphi \approx 1$ to fit the condition of $\gamma_{12} \ll \gamma_1$ or $\gamma_2$[65].

### Mapping the phase diagram of the CNC-PEG-dextran mixtures

The incorporation of CNCs significantly alters the critical concentration of polymers necessary for LLPS transition and governs transitions from different phase regions. To begin, the critical composition for LLPS transition in the four-component mixtures were examined. The dextran concentration was fixed between 3 and 5 wt% (stepwise 0.25 wt%), and the PEG concentration was varied to identify the critical composition for LLPS transition at different CNC concentrations. A specific quantity of Milli-Q water was added to ensure a consistent total mass of 4 g for the four-component mixtures. When operating at either 21 or 50 °C, the critical points in the binary polymer solution of PEG and dextran served as the baseline. With the incorporation of 1 wt% of CNCs, The PEG concentration was then gradually reduced in a stepwise manner by 0.05 wt% from the baseline critical concentration at each dextran concentration. This process continued until the equilibrated four-component mixtures no longer exhibited an LLPS transition, which identified the new critical PEG-dextran composition for LLPS transition at 1 wt% of CNCs. This methodology was then replicated at each incremental CNC concentration to determine the corresponding critical polymer compositions. Notably, the critical concentration of CNC for LCPS to occur was identified when the cholesteric phase

became apparent in the equilibrated samples (verified between crossed polarizers).

At 50 °C, when the CNC concentration reached 4.5 wt%, inverse-phase separation phenomenon (a cholesteric phase on top of an isotropic phase) was observed, occurring at the critical PEG-dextran compositions for LLPS transition. As the PEG content increased, the system transitioned from an inverse-phase state to a four-phase state. To determine the phase boundary separating the inverse-phase from the four-phase regions, the PEG concentration was systematically increased (stepwise 0.05%) from the critical PEG concentration at each specific dextran concentration, until the mixtures displayed four phases. Finally, gathered data points were processed using the software MATLAB 2023a to construct a three-dimensional phase diagram.

## Data availability

The data that support the findings of this study are available in the main text or the supplementary materials, or available from the corresponding authors upon request. Source data are provided with this paper.

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

## Acknowledgements

G.C. acknowledges the financial support from the Novo Nordisk Foundation (NNF20OC0064350). O.J.R. and G.C. acknowledge the financial support from the Canada Excellence Research Chair initiative (CERC-2018-00006), the European Research Council under the European Union's Horizon 2020 research and innovation program (ERC Advanced grant No. 788489, "BioElCell"), and the Canada Foundation for Innovation (Project number: 38623). Han acknowledges financial support from China Scholarship Council (CSC No. 202006950015) and Academy of Finland (Nos. 318890 and 318891, Competence Centre for Materials Bioeconomy, FinnCERES). Han acknowledges the provision of facilities and technical support by Aalto University at OtaNano-Nanomicroscopy Center (Aalto-NMC) and the assistance of Dr. Wenwen Fang for viscosity measurements. Sincere gratitude goes to Dr. Daniel de las Heras and Tobias Eckert for valuable discussion.

## Author contributions

H.T. prepared aqueous CNC-PEG-dextran mixtures and carried out the experimental measurements and data analysis. C.R., H.L., J.Z., A.K. and J.T.: data curation. O.J.R. and E.K.: funding acquisition, writing and review. G.C. designed and led the project, funding acquisition, formal analysis, writing and review. The manuscript was written through contributions of all authors. All authors have given approval to the final version of the manuscript.

## Competing interests

The authors declare no competing interests.
