## [Peer Review File · Nature Communications]

Thermodynamically Controlled Multiphase Separation of Heterogeneous Liquid Crystal ColloidsREVIEWER COMMENTS

Reviewer #1 (Remarks to the Author):

In lines 75-78, page 5, the authors wrote: "In this study, we report a thermodynamically controlled multiphase separation with cholesteric liquid crystal assembly in a three-component soft matter system. Specifically, the system consists of an aqueous mixture of cellulose nanocrystals (CNCs), poly (ethylene glycol)(PEG) and dextran." Water is the main constituent of the system. Why do the authors only consider the CNCs, PEG, and Dextran as the system's main components?

In lines 126 to 128, page 7, "When the cholesteric CNC-polymer mixtures were heated, both the anisotropic volume fraction and the helical pitch of the resulting phase slightly decreased, attributed to the flexibility of the polymer chains, and the expanded polymer conformation" The explanation why the liquid crystalline phase and the pitch decrease is not convincing, it is challenging to understand.

Lines 187-188, page 10 "To quantify the interplay between LLPS and LCPS, we fixed the CNC-PEG concentration (6 wt%-3.75 wt%) and tuned the dextran concentration varied across two critical polymer..." The selection of the range of concentration for CNC-PEG should be better explained.

Lines 206 to 208, page 11 "The measured helical pitch in the bulk phase of the cholesteric PEG-rich region was larger than that of the dextran-rich region, implying a grading pitch along the polymer variation." Why this variation of the pitch?

The authors refer to the viscoelasticity of the system. However, only the viscosity of the PEG-dextran system with the composition 3.75 wt%-4,25wt% was measured (Newtonian fluid). The other rheological properties still need to be addressed.

The system presented by the authors is interesting. However, many parameters are involved, including the concentration of the CNCs, dextran, and PEG in water and the temperature, which make the system very complicated to be understood.

Reviewer #2 (Remarks to the Author):

The authors have made an extensive study of a series of 4-component colloids consisting of aqueous dispersions of cellulose nanocrystals (CNCs) as well as the polymers polyethylene glycol (PEG) and Dextran dissolved in the water, uncovering some quite spectacular phenomena, such as phase separation between multiple cholesteric liquid crystal phases and unexpected density differences, leading to liquid crystal phases floating on top of isotropic phases in the gravitational field. The system is extraordinarily complex, requiring extreme care in the analysis and a flawless understanding of a range of phenomena, from depletion attraction to liquid crystal formation to phase separation. Unfortunately this is where the manuscript fails, as the authors demonstrate what appears to be a number of deficiencies in their understanding, if not misunderstandings, sometimes on rather serious issues. This means that the analysis fails at a fundamental level, yielding a manuscript that is speculative in character. It is also often very difficult to follow the logic and understand the reasoning. I assume that the latter is a consequence of the former. Thus, while the system is interesting and the experiments intriguing, the authors need to start by studying the basics of the issues at discussion, and then redo the analysis from scratch, making an effort to be clear and quantitative in the discussion and descriptions. I cannot go through all details, but here are the most critical problems.

1. It is apparent from the authors' 'artistic' drawings of how the CNCs organize in their systems that they have a flawed idea of the type of order prevailing in cholesterics. There are no layers in a cholesteric. A cholesteric is a chiral nematic phase and therefore, just like for a non-chiral nematic, there is no long-range positional order, only orientational. A good illustration of what the cholesteric organization of nanorods looks like can be found in Fig. 4a in "Entropy-driven formation of chiral nematic phases by computer simulations", S. Dussi and M. Dijkstra, *Nat. Commun.*, 7:11175 (2016). As is clear from this figure, there is no separation of CNCs into layers like the authors have drawn in their Fig. 2b, Fig. 3g, and Fig. 5g,h. The fact that SEM images of cross sections reveals a stratified character is not an evidence of layering, but only a result of the helically modulated mechanical properties. This is well known and has been discussed, for instance, in "From Equilibrium Liquid Crystal Formation and Kinetic Arrest to Photonic Bandgap Films Using Suspensions of Cellulose Nanocrystals" by Schütz et al., *Crystals*, 10:3, p.199 (2020).

2. This misunderstanding then feeds into a skewed idea of how depletion attraction affects the behavior of the system, as it appears that the authors believe the depletants go in between the non-existent layers. Even if there were layers, as in the case of smectic organization, the depletants would not go in between layers, but they rather stabilize individual layers or filaments of smectic organization, since the confinement of the depletants between layers would come at a great entropy cost. This phenomenon was actually studied in suspensions of monodisperse fd-virus suspensions subject to depletion attraction induced by Dextran addition, see e.g. “Entropy driven self-assembly of nonamphiphilic colloidal membranes” by E. Barry and Z. Dogic, *Proc. Natl. Acad. Sci. U.S.A.*, 107:23, p. 10348 (2010), “Self-assembly of 2D membranes from mixtures of hard rods and depleting polymers” by Y. Yang et al., *Soft Matter*. 8:3, p.707 (2012) and “Reconfigurable self-assembly through chiral control of interfacial tension” by T. Gibaud et al., *Nature*, 481:7381, p. 348 (2012). In contrast to viruses, CNCs are notoriously polydisperse and therefore do not form smectic structures. For CNCs, the Dextran addition would normally be expected to condense the CNCs into tactoids of high CNC concentration without any incorporated Dextran, as in Fig. 1B in the *Soft Matter* paper by Yang et al. This is because there is no entropic gain for the Dextran molecules to be within a condensed liquid crystal phase. The effect of depletion attraction must instead be to reduce the global concentration of CNC required to form a liquid crystal phase, because within the cholesteric nuclei, the CNC concentration is much higher than outside, thanks to the depletants. I would strongly recommend the authors to read the seminal work by Dogic and co-workers and discuss their own findings in that context. In fact, an intriguing possibility is that the authors of this work have induced fractionation of the CNC by the action of depletants, such that smectic membranes might form even in the CNC system. I doubt that this is the case, but it is not impossible. If evidence for such an effect would be found, that would indeed be spectacular. But testing this requires a proper analysis far beyond what is presented in this manuscript.

3. While it is perfectly acceptable with English mistakes in manuscripts written by authors who do not have English as their mother tongue, the type of formulation mistakes in this manuscript make me concerned that there is more than language issues at play here. I cannot help wonder if the authors have fully understood the essence of key concepts.

Already in the abstract (line 23), the statement that a system were to consist of different types of phase separations raises a red flag. Phase separation is a phenomenon; something is happening. A system cannot consist of phase separation. In the third sentence of the introduction (line 41), the authors then write that a colloid may phase separate in order to "gain" free energy; it is sophomore level thermodynamics that any spontaneous process takes place in the direction of reduced free energy, not to gain higher free energy. In the second paragraph, the authors get into a mode of extreme generalization, for instance suggesting (line 54) that liquid-liquid phase separation (LLPS) would require the presence of macromolecules and water; neither is required, as LLPS is very general, with many examples in mixtures of organic small molecules, without any water.

4. Fig. 1, quite central to the discussion throughout the paper, is full of ambiguity and/or errors. Panel (a) is said to show a "volume-composition diagram" but there is no volume axis in this diagram; it is a 1D diagram, the only variable being composition, increasing from left to right. The diagram in panel (b) is incomprehensible: What is the black curve? What are the coils drawn in the region labelled "Two phases"? Are there no polymer coils in the single-phase system? If the black curve is a binodal coexistence curve, there are no states at all in the middle, between the two flanks below the plait point. Where is the temperature or pressure that are mentioned in the text (line 55) that refers to this diagram? What are the compositions of the two coexisting phases? What happens above the peak of the black curve; are the authors suggesting that there is only one phase at high enough concentration? Why is then the bottom, low-concentration, region labelled as "One phase"? In the caption to panel (c), the authors write that the two lines are coexistence lines, but that cannot be true because such lines indicate which two phases coexist; there is no such indication here. Coexistence lines are lens- or bell-shaped (similar to the black curve in panel (b)). Moreover, what is the physical origin of these lines in panel (c)? Or are they simply hand-drawn? The caption also refers to "the binodal curve" without such a curve ever having been introduced; where is it? A binodal curve is bell-shaped with a plait point at the top or at the bottom. I see no such curve in panel (c).

5. The confusion continues on page 5, for instance with the idea (line 85) that cooling would "shift" a binodal phase diagram toward lower concentration of the dissolved polymers PEG

and Dextran; what they are doing is shifting their system into a different part of the phase diagram. The same problem reappears on page 8, line 145 in the text “the phase diagram of CNC-PEG-dextran mixtures had shifted in the favor of a two-phase over a one-phase system”. The phase diagram does not shift; the authors are shifting their sample composition such that it ends up in a different part of the four-component phase diagram. It is hardly surprising that you see biphasic regimes as a result of this. To understand what is going on, in particular in the following experiments where the CNC fraction increases, it would be important to know what the phase diagram looks like and what the compositions of the different phases are. This is actually a key problem with this paper: they never bother to even sketch a phase diagram or to clearly define how each phase is composed. Without some kind of diagrams explaining what is going on as compositions change in their 4-component system (that’s another mistake on page 5, line 76; they write that it is a 3-component system, apparently forgetting the solvent, which is extremely important for the behavior), every discussion of different phases remains extremely vague. Admittedly, representing 4-component phase diagrams is not easy, and obviously requires defining a set of representative 2- or 3-component phase diagrams, where the other one or two components are kept constant (as well as temperature, which adds a fifth variable to the study, since they also vary temperature). But it was the authors’ choice to go for such a complex system; they need to break it down into pieces that can be represented, conducting their experiments accordingly and referring to each 2- or 3-dimensional diagram in analyzing the outcomes.

6. An example of a situation where an understanding of the compositions of the separating phases is critical to a correct interpretation is found on page 7, line 127, where the authors conclude that liquid crystal phase separation (LCPS) could be triggered “not only by the concentration, but also modified by adding semiflexible nonadsorbing polymers”. However, as is clear from the above-mentioned works by Dogic and co-workers, the impact of depletion attraction on a colloidal system developing LC phases is to nucleate phases with much higher local concentration of the particles than in the global system. In other words, since the authors do not take the different concentration of CNC in the separating LC phase into account, they incorrectly conclude that concentration would not play a role in the liquid crystal phase separation.

7. The idea to contrast LLPS against LCPS is not necessarily appropriate. On page 8, line 147, they write that “both LLPS of polymers and LCPS of CNCs occurred in the hybrid system”, but it is not correct to attribute LLPS to polymers and LCPS to CNCs. All components influence the behavior of the system. The authors demonstrated just before that the addition of a small amount of CNCs can induce LLPS without any liquid crystal formation. Likewise, the presence of polymers will influence the liquid crystal formation and related phase separation. With two types of polymers dissolved into the solvent of a polydisperse nanoparticle suspension, they have created a very complex system, and they have to consider the impact of every component on every phenomenon.

On page 18, line 318, the sentence “As the multiphase separation continues, LLPS reach equilibrium and LCPS begin” is problematic. This would suggest that the initial LLPS gives rise only to isotropic phases, and only when the “LCPS begin”s, an anisotropic liquid crystalline phase would be seen. However, the long paragraph preceding this sentence, supposedly describing LLPS, refers to Fig. S18–S21, which all clearly reveal the presence of liquid crystal phases already at this stage. The DIC images in Fig. S18 and S21 have many areas with fingerprint lines typical of cholesteric phase formation, and the POM images in Fig. S19-20 are very colorful, clearly demonstrating that the phases are birefringent. The idea that first LLPS would complete and reach equilibrium, and only after this LCPS would take place is thus contradicted by the experimental evidence.

8. Considering the importance of PEG and Dextran in this study, I am surprised that the authors never discuss why a ternary mixture of PEG and Dextran in water will tend to separate at certain temperatures but stay homogeneous at others, and why their interaction with CNCs is different. I imagine that the sugar ring structures in Dextran give it a more favorable interaction with CNC than PEG, but the authors never discuss this. Likewise, I cannot find the evidence for their statements of a certain phase being PEG-rich and another being Dextran-rich; they just write, for instance on line 149, that the bottom phase is Dextran-rich and the upper is PEG-rich, but where is the experimental evidence? This question recurred throughout the manuscript while I was reading.

9. On page 11, lines 204–206, the authors write that a fingerprint texture of one phase is

“continuously invaded” from another cholesteric phase, without presenting any picture evidence. It is unclear what is meant by this. They then go on by describing the interface as being an “immiscible PEG-dextran interface”. What is meant by “immiscible” here? An interface can neither be miscible nor immiscible. Do they mean that the two sides of the interface contain only PEG and Dextran, respectively? That will hardly be the case, as phase separation never leads to separation of a pure component, unless that component is crystallizing out as a solid crystal, which is not the case here. The confusion regarding the separation phase compositions gets rather extreme at some points, like line 297-298 on page 16, where the authors write that LLPS-induced nuclei would be “polymer droplets”; of course they are not pure droplets of polymer, but probably a liquid solution enriched in one or both polymers.

10. Also the descriptions of data are incomplete. For instance, to the many images of samples in which phase separation occurs, the authors write in the caption only that they are “photographs”. How were they taken? Are the samples between crossed polarizers? If not, how do the authors distinguish between liquid crystalline and isotropic phases?

Reviewer #3 (Remarks to the Author):

This paper described a controllable phase transition coupled with temperature-dependent liquid-liquid phase separation (LLPS) and concentration-dependent liquid crystal phase separation (LCPS). The observation of the assembly process of cellulose nanocrystals (CNCs) in the compartmentalized polymer regions appeared convincing. The analysis of the influencing factors of the phase behavior was done with great care. The conclusions were drawn based on appropriate controls and sufficiently strong evidence. The list of references was comprehensive and well placed. The paper was, in most parts, pleasant to read and the reasoning was easy to follow. It presented an advancement in the understanding of the underlying science of the similar systems (their early publication in ACS Nano).

Having said that, the work would be more appealing if a proof-of-concept illustration can be presented to showcase the applications of their studies. The followings are a few technical issues to be addressed:

1. Wordings in Figure 3 were too small to read.
2. Scale bars in Figures 3-5 were partially labelled.
3. 'SAXS' in Figure S16 was miss-spelt.
4. In line 58 of SI, 'pith' was miss-spelt.

I recommend acceptance for publication after satisfactory addressing of the above.

Referee 1:

1. In lines 75-78, page 5, the authors wrote: “In this study, we report a thermodynamically controlled multiphase separation with cholesteric liquid crystal assembly in a three-component soft matter system. Specifically, the system consists of an aqueous mixture of cellulose nanocrystals (CNCs), poly (ethylene glycol) (PEG) and dextran.” Water is the main constituent of the system. Why do the authors only consider the CNCs, PEG, and Dextran as the system's main components?

Response: We appreciate the Reviewer's input on this issue. We aim to investigate the multiphase separation in aqueous CNC-PEG-dextran mixture which consist of LCPS of CNCs and LLPS of polymers. In this context, water is the background medium that disperses the components that undergo phase separation. We agree with the Reviewer that water plays a pivotal role in the multiphase system. Water can not only act as the dispersion medium for colloids, but also impact the balance between enthalpic and entropic factors in the phase separation process. For example, the LLPS transition process can be strongly influenced by tuning polymer-water interaction through temperature adjustments.

Based on the Reviewer's comment, we highlight the contribution of water during the phase separation and indicate the heterogeneous system as four-component soft matter.

2. In lines 126 to 128, page 7, “When the cholesteric CNC-polymer mixtures were heated, both the anisotropic volume fraction and the helical pitch of the resulting phase slightly decreased, attributed to the flexibility of the polymer chains, and the expanded polymer conformation” The explanation why the liquid crystalline phase and the pitch decrease is not convincing, it is challenging to understand.

Response: We thank the Reviewer for raising this issue. In CNC-polymer mixtures, the depletion interaction between nanoparticles and non-adsorbing polymers forms an exclusion layer around CNC, which prevents polymer adsorption. As neighboring CNC approach, their corresponding depletion zones overlap and create an imbalance in the osmotic pressure exerted on each nanoparticle. This leads to an attractive depletion interaction that is affected by the size of the polymer and concentration (*Journal of Polymer Science*, **1958**, 33, 183–192.). Upon heating, the hydrodynamic size of PEG and dextran slightly increases due to the expansion of the polymer chains (**Figure 1**). When CNC-polymer mixtures are heated, the attractive depletion interactions between CNCs are altered (by changes of neighboring polymer chains), resulting in a decreased anisotropic volume fraction and helical pitch of the liquid crystal phase.

Figure 1. Dynamic light scattering of (a) CNC, (b) PEG and (c) dextran aqueous suspension, respectively. These results demonstrate that the hydrodynamic size of PEG and dextran coils increased with temperature, whereas CNC remains constant.

The relevant discussion in the revised manuscript has been rephrased as:

“Owing to the flexibility and expanded conformation of the polymer chains, when the cholesteric CNC-polymer mixtures are heated, both the anisotropic volume fraction and the helical pitch of the resulting cholesteric phase slightly decreases, which can be attributed to the changes of attractive depletion interactions between CNCs (Figure 2c, Figure S6, and Table S1).”

3. Lines 187-188, page 10 “To quantify the interplay between LLPS and LCPS, we fixed the CNC-PEG concentration (6 wt%-3.75 wt%) and tuned the dextran concentration varied across two critical polymer concentrations.” The selection of the range of concentration for CNC-PEG should be better explained.

Response: We appreciate the Reviewer for highlight this concern. The selected concentration range (CNC-PEG, 6 wt%-3.75 wt%) was derived from the experimental observations. **Figure 2** shows that when the concentration of the neat CNC suspension reaches 6 wt%, a cholesteric liquid crystal phase forms without the absence of an isotropic phase. This makes the 6 wt% sample an ideal starting point to investigate the influence of non-absorbing polymers on LCPS of CNCs.

In the PEG-dextran binary polymer solution at 21 °C (**Figure 3**), we noted that by fixing the PEG concentration at 3.75 wt% and fine-tuning the dextran concentration (from 3.5 to 4.5 wt%) makes it possible for the binary polymer solution to transition from a miscible state to a two-phase system through LLPS. However, within the selected dextran range (4.25 wt%, 4.50 wt%), the binary polymer solution remains homogeneous under 50 °C and separating into two phases at 21 °C. When the CNC-PEG concentration was fixed at 6 wt%-3.75 wt%, adjusting the dextran concentration can not only examine the influence of polymers on the LCPS of CNCs, but also check the LLPS phase behavior of PEG-dextran with the addition of CNC. In essence, this approach allows us to better understand the interplay between LLPS and LCPS.

Figure 2. Photograph of aqueous CNC suspensions to highlight the concentration dependent liquid crystal phase separation (the samples were placed between crossed polarizers).

Figure 3. Experimental phase diagram of binary polymer solutions of PEG (8 kDa) and dextran (450-600 kDa) at 21 and 50 °C. The coexistence lines (black and red) separate the one-phase (below the lines) and two-phase regimes (above the lines) as a function of environmental temperature.

4. Lines 206 to 208, page 11 “The measured helical pitch in the bulk phase of the cholesteric PEG-rich region was larger than that of the dextran-rich region, implying a grading pitch along the polymer variation.” Why this variation of the pitch?

Response: We apologize for the lack of clarity regarding this issue. The helical pitch of cholesteric liquid crystal phase is relevant to the distance between neighboring CNC nanorods, which is influenced by the depletion mechanism in CNC-polymer mixtures. The strength of depletion interaction is sensitive to the concentration of nonadsorbing polymers (*Journal of Polymer Science*, **1958**, 33. 183–192). According to the size-exclusion chromatography results (see **Table 1**), the concentration of PEG and dextran varies across each phase. Therefore, the differences of depletion interactions acting on the CNC nanorods result in the variations of cholesteric helical pitch between PEG-rich and dextran-rich regions.

Based on the Reviewer’s comment, the relevant discussion in the revised manuscript has been rephrased as:

“The measured helical pitch in the bulk phase of the cholesteric PEG-rich region was larger than that of the dextran-rich region, implying a grading pitch along the polymer concentration variation which due to the differences of depletion interactions (Table S2).”

Table 1 Relative composition of each phase in the equilibrated CNC-PEG-dextran aqueous mixtures based on size-exclusion chromatography data.

	Initial Composition (wt%)			Density (g/cm ³)	Concentration (mg/mL)		
	PEG	Dextran	CNC		PEG	Dextran	
21 °C	3.75	4.25	0	PEG-rich phase	1.025	40.82	31.36
				Dextran-rich phase	1.063	21.23	72.99
	3.75	3.5	6	PEG-rich isotropic phase	1.025	42.23	17.85
				PEG-rich cholesteric phase	1.175	50.00	13.57
				Dextran-rich cholesteric phase	1.242	17.13	81.20
				Dextran-rich isotropic phase	1.032	44.90	8.28
3.75	4.25	6	PEG-rich cholesteric phase	1.070	50.38	11.11	
			Dextran-rich cholesteric phase	1.238	14.22	96.68	
			Dextran-rich isotropic phase	1.193	43.23	26.83	
			Dextran-rich isotropic phase	1.234	21.98	66.88	
50 °C	3.75	3.5	6	PEG-rich isotropic phase	1.138	56.43	14.58
				PEG-rich cholesteric phase	1.208	54.36	15.75
	3.75	4.25	6	Dextran-rich isotropic phase	1.235	25.18	70.46
				Dextran-rich cholesteric phase	1.270	21.97	75.67

5. The authors refer to the viscoelasticity of the system. However, only the viscosity of the PEG-dextran system with the composition 3.75 wt%-4.25wt% was measured (Newtonian fluid). The other rheological properties still need to be addressed.

Response: We thank the Reviewer for the valuable suggestions. We have performed additional rheology measurements to determine the viscosity profile and viscoelastic behavior of each phase in the PEG-dextran system and equilibrated CNC-PEG-dextran mixtures at both 21°C and 50°C (**Figure 4**).

Based on the Reviewer's comment, **Figure 4** has been added into the revised Supporting Information as Figure S24.

Figure 4. (a)-(c) Viscosity profile and viscoelastic behavior (storage modulus G' with filled symbols and loss modulus G'' with open symbols) for the PEG-dextran system (3.75 wt%-4.25 wt%) and CNC-PEG-dextran aqueous mixture (6.0 wt%-3.75 wt%-4.25 wt%) at different temperature.

6. The system presented by the authors is interesting. However, many parameters are involved, including the concentration of the CNCs, dextran, and PEG in water and the temperature, which make the system very complicated to be understood.

Response: We appreciate the Reviewer for recognize the novelty of our work. The current multiphase separation system can be conceptualized as LCPS of CNCs and LLPS of PEG and dextran. The phase behavior is due to the interplay between the pairs of phase transition processes. To better present the interplay between LCPS and LLPS, we have invested considerable efforts to map the three-dimensional phase diagram of the aqueous CNC-PEG-dextran mixtures at different temperatures (**Figure 5**). The phase diagrams are introduced as a framework to rationalize the influence of CNCs, PEG and dextran on the phase behavior. We hope that the Reviewers and the readers find the added information effective to better understand the multiphase separation system.

Figure 5. Three-dimensional phase diagram of the CNC-PEG-dextran aqueous mixture that highlight the interplay between LCPS of CNCs and LLPS of polymers at 21 °C (a) and 50 °C (b), respectively. Among which, the LLPS process is inhibited at the left side of the CNC-PEG-dextran coexistence plane (gray dots) whereas the right side area is active for LLPS. With the contribution of LCPS, the resulting multiphase regions are modulated by the critical CNC concentrations (colored horizontal planes). (c) When the CNC concentration exceeds 4.5 wt%, the multiphase transition area can be defined as two-phase region (only LCPS, points 1 and 4), inverted phase region (where a cholesteric phase is on top of an isotropic phase, points 2 and 5), and a four-phase region (points 3 and 6) due to changes of the polymer composition.

Based on the Reviewer's comment, **Figure 5** has been added into the revised manuscript along the relevant discussion:

“To better understand the multiphase separation, we clarify the interplay between LLPS and LCPS by mapping the three-dimensional phase diagrams for the four-component CNC-PEG-dextran aqueous mixtures at 21 and 50 °C, respectively (Figure 3). When the CNC concentration is below 3.25 wt% and equilibrate under 21 °C, the phase diagram of CNC-PEG-dextran aqueous mixture can be divided into one-phase region and two-phase region by the coexistence plane (gray dots). The obtained mixtures remain homogeneous with the compositions located at the left side of the CNC-PEG-dextran coexistence plane and phase separated due to the contribution of LLPS process at a composition on the right side of the coexistence plane (Figure 3a). The phase separation behavior becomes more pronounced by increasing the CNC concentration, which is ascribed to the occurrence of LCPS. Given the multiphase separation process, the obtained CNC-PEG-dextran aqueous mixture display either a two-phase or three-phase behavior. It should be noted that the origin of the two-phase behavior can be realized by the contribution of LCPS or coupled LLPS-LCPS. Further increasing CNC concentration to 5.5 wt% leads to the emergence of an additional cholesteric phase within the PEG-rich region, thereby transforming the hybrid system from a two-phase to a three-phase state.

Upon heating, the interactions of CNC-polymers and PEG-dextran changed, leading to temperature-sensitive multiphase separation with two-, three- and four-phase coexistence behaviors of different physical origins (Figure 3b). To explore the multiphase separation under 50 °C, the three-dimensional phase diagram can be partitioned by the CNC-PEG-dextran coexistence plane and the critical CNC concentration planes into several regions. With the addition of polymers, the LCPS of CNC is profoundly suppressed; namely, the CNC concentration threshold for LCPS shifts from 3.25 wt% (21 °C) to 4.50 wt% (50 °C) in the polymer miscible region. However, in the region where LLPS occurs, the critical concentration for LCPS of CNC remains at 3.25 wt%. Above this critical CNC concentration, the four-component CNC-PEG-dextran aqueous mixtures demonstrate three-phase behavior with LLPS and LCPS process, including an upper PEG-rich isotropic phase, a middle dextran-rich isotropic phase, and a bottom dextran-rich cholesteric phase. Intriguingly, when the CNC concentration reaches 4.50 wt%, the resulting mixtures exhibit more complex phase behavior with the emergence of varying inverted cholesteric-isotropic states. Figure 3c highlight the main critical coexistence plane (gray dots) and secondary phase plane (magenta square) with a transition from two-phase, three-phase to four-phase states, where the intermediate region between the two phase boundaries displays a unique inverted phase behavior. Within this region, the volume fraction of upper cholesteric PEG-rich phase increases with the increasing of CNC concentration, and finally the multiphase system shifts from three-phase to two-phase stacking (from point 5 to 2). Further changes in the polymer concentration, below the main critical coexistence plane or beyond the secondary phase plane, leads to two-phase (point 1 and 4) and four-phase behaviors (point 3 and 6), respectively.”

Referee 2:

1. The authors have made an extensive study of a series of 4-component colloids consisting of aqueous dispersions of cellulose nanocrystals (CNCs) as well as the polymers polyethylene glycol (PEG) and Dextran dissolved in the water, uncovering some quite spectacular phenomena, such as phase separation between multiple cholesteric liquid crystal phases and unexpected density differences, leading to liquid crystal phases floating on top of isotropic phases in the gravitational field. The system is extraordinarily complex, requiring extreme care in the analysis and a flawless understanding of a range of phenomena, from depletion attraction to liquid crystal formation to phase separation.

Response: We appreciate the opportunity to revise the manuscript. We are thankful for the extensive and relevant feedback provided by the Reviewer which we found very useful.

2. Unfortunately this is where the manuscript fails, as the authors demonstrate what appears to be a number of deficiencies in their understanding, if not misunderstandings, sometimes on rather serious issues. This means that the analysis fails at a fundamental level, yielding a manuscript that is speculative in character. It is also often very difficult to follow the logic and understand the reasoning. I assume that the latter is a consequence of the former. Thus, while the system is interesting and the experiments intriguing, the authors need to start by studying the basics of the issues at discussion, and then redo the analysis from scratch, making an effort to be clear and quantitative in the discussion and descriptions. I cannot go through all details, but here are the most critical problems.

Response: We have revised the manuscript to address the points raised by the reviewer and considering the suggestions provided by the other reviewers. Additional experiments were carried out to discuss the basic issues and to more effectively analyze and present the result. The manuscript has been considerably improved and provides a more clear and quantitative discussion.

3. It is apparent from the authors' 'artistic' drawings of how the CNCs organize in their systems that they have a flawed idea of the type of order prevailing in cholesterics. There are no layers in a cholesteric. A cholesteric is a chiral nematic phase and therefore, just like for a non-chiral nematic, there is no long-range positional order, only orientational. A good illustration of what the cholesteric organization of nanorods looks like can be found in Fig. 4a in "Entropy-driven formation of chiral nematic phases by computer simulations", S. Dussi and M. Dijkstra, *Nat. Commun.*, 7:11175 (2016). As is clear from this figure, there is no separation of CNCs into layers like the authors have drawn in their Fig. 2b, Fig. 3g, and Fig. 5g,h. The fact that SEM images of cross sections reveals a stratified character is not evidence of layering, but only a result of the helically modulated mechanical properties. This is well known and has been discussed, for instance, in "From Equilibrium Liquid Crystal Formation and Kinetic Arrest to Photonic Bandgap Films Using Suspensions of Cellulose Nanocrystals" by Schütz et al., *Crystals*, 10:3, p.199 (2020).

Response: We thank the Reviewer for this insightful comment to point out our misunderstanding related to the cholesteric organized liquid crystal. Based on the suggestions provided by the Reviewer, we have corrected the corresponding illustrations to remove the "layer structure" in revised manuscript (see **Figure 6-9** in Response and Figure 2b, Figure 4g, Figure 6g, h in the revised manuscript). These figures accurately describe the long-range orientational order that reflects how the CNCs self-assemble in the polymer solutions.

Figure 6. Illustration of cholesteric self-assembly of CNC with nonadsorbing polymers through depletion effect.

Figure 7. Schematic illustration of the PEG-dextran interface with cholesteric-cholesteric (left) and cholesteric-isotropic (right) stacking sequence across a gradient of polymer concentration.

Figure 8. A schematic description of the coupled LLPS-LCPS process through nucleation and growth of polymers inside the tactoids.

Figure 9. Illustration of the inversed multiphase separation due to the accumulation of negative and positive tactoids.

4. This misunderstanding then feeds into a skewed idea of how depletion attraction affects the behavior of the system, as it appears that the authors believe the depletants go in between the non-existent layers. Even if there were layers, as in the case of smectic organization, the depletants would not go in between layers, but they rather stabilize individual layers or filaments of smectic organization, since the confinement of the depletants between layers would come at a great entropy cost. This phenomenon was actually studied in suspensions of monodisperse fd-virus suspensions subject to depletion attraction induced by Dextran addition, see e.g. “Entropy driven self-assembly of nonamphiphilic colloidal membranes” by E. Barry and Z. Dogic, *Proc. Natl. Acad. Sci. U.S.A.*, 107:23, p. 10348 (2010), “Self-assembly of 2D membranes from mixtures of hard rods and depleting polymers” by Y. Yang et al., *Soft Matter*, 8:3, p.707 (2012) and “Reconfigurable self-assembly through chiral control of interfacial tension” by T. Gibaud et al., *Nature*, 481:7381, p. 348 (2012). In contrast to viruses, CNCs are notoriously polydisperse and therefore do not form smectic structures. For CNCs, the Dextran addition would normally be expected to condense the CNCs into tactoids of high CNC concentration without any incorporated Dextran, as in Fig. 1B in the *Soft Matter* paper by Yang et al. This is because there is no entropic gain for the Dextran molecules to be within a condensed liquid crystal phase. The effect of depletion attraction must instead be to reduce the global concentration of CNC required to form a liquid crystal phase, because within the cholesteric nuclei, the CNC concentration is much higher than outside, thanks to the depletants. I would strongly recommend the authors to read the seminal work by Dogic and co-workers and discuss their own findings in that context. In fact, an intriguing possibility is that the authors of this work have induced fractionation of the CNC by the action of depletants, such that smectic membranes might form even in the CNC system. I doubt that this is the case, but it is not impossible. If evidence for such an effect would be found, that would indeed be spectacular. But testing this requires a proper analysis far beyond what is presented in this manuscript.

Response: We sincerely appreciate the valuable feedback volunteered by the reviewer, along the reference and the suggestions. In order to confirm the Reviewer's concern regarding to the depletion behavior in CNC-dextran system, we have conducted an additional experiment using FITC-dextran (0.01wt%) as fluorescent tracer in the CNC-dextran (6 wt%-4.25 wt%) mixture to verify the co-assembly behavior between CNC and dextran macromolecules. The obtained results demonstrate strong fluorescent signals both in the cholesteric tactoid and its isotropic surroundings, suggesting that dextran molecules can be incorporated into the anisotropic phase of CNCs (see **Figure 10** and Figure S3 in the revised manuscript). Our results are consistent with previous studies that shown the compatibility of CNC liquid crystal phase with dextran coils (*Macromolecules* **2002**, 35, 7400–7406; *Macromolecules* **2007**, 40, 3429–3436). Based on our experimental results, the distance between neighboring CNCs (~ 37 nm calculated by SAXS data) in the cholesteric phase is approximately three times larger than the size of dextran coils (hydrodynamic size ~ 10 nm), which points to the possibility for dextran coils to be assembled in the cholesteric phase between neighboring CNC, even at some extent.

Figure 10. Fluorescence image of the CNC-dextran mixture (6 wt%-4.25 wt%) with 0.01 wt% FITC-dextran as tracer, revealing strong fluorescent signals both in the cholesteric tactoid and its surrounding isotropic phase. These results suggest that dextran molecules were incorporated into the cholesteric liquid crystal phase of CNCs.

Inspired by the Reviewer's comments, we further check the depletion differences between PEG and dextran in CNC suspension with the concentration of 6 wt%-4.25 wt% (CNC-polymer). Both mixtures displayed a large area of long-range ordered fingerprint texture, typical of the cholesteric organization of CNCs (see **Figure 11** and Figure S4 in the revised manuscript). For the anisotropic CNC-PEG mixture, we observed a vesicle-like structure with isotropic interior and cholesteric surroundings (**Figure 11a**). However, the anisotropic phase of CNC-dextran mixture exhibited homogenous fingerprint texture with minimal disclinations, and notably, without the formation of vesicle-like structures (**Figure 11b**). These results indicate that the two polymer depletants exhibit different interactions with CNCs, which reflect the higher affinity that exist between CNC and dextran compared to that with PEG. One possibility is that the addition of PEG may induce the assembly of similarly sized CNC nanoparticles into smectic organization within the cholesteric phase, as suggested by the Reviewer. However, detailed exploration of this intriguing hypothesis needs to be pursued in future research.

Figure 11. POM images of the CNC-PEG (6 wt%-4.25 wt%) (a) and CNC-dextran (6 wt%-4.25 wt%) (b) mixtures, displaying a distinctive fingerprint texture typical of cholesteric organization. Specifically, vesicle-like structures are observed in the CNC-PEG mixture with isotropic interior and cholesteric surroundings, while the CNC-dextran mixture is homogeneous with long-range ordered fingerprint texture.

We have carefully read the references recommended by the Reviewer and better realize the complexity of polymer depletion induced self-assembly of rod-like nanoparticles. These works have enriched our understanding the interplay between hard rod-like nanoparticles and non-absorbing polymers through depletion mechanisms. Therefore, we have cited these the references in the revised manuscript.

The relevant discussion in the revised manuscript has been rephrased as:

“Owing to the molecular differences between PEG and dextran, the microscopic texture of CNC-PEG and CNC-dextran liquid crystal phase are slightly different under depletion effect (Figure S4). This maybe because of dextran (composed by branched glucose units) share similar molecular structure with CNCs (composed by linear chain glucose units), leading to more favorable steric and intermolecular interactions.”

5. While it is perfectly acceptable with English mistakes in manuscripts written by authors who do not have English as their mother tongue, the type of formulation mistakes in this manuscript make me concerned that there is more than language issues at play here. I cannot help wonder if the authors have fully understood the essence of key concepts. Already in the abstract (line 23), the statement that a system were to consist of different types of phase separations raises a red flag. Phase separation is a phenomenon; something is happening. A system cannot consist of phase separation. In the third sentence of the introduction (line 41), the authors then write that a colloid may phase separate in order to “gain” free energy; it is sophomore level thermodynamics that any spontaneous process takes place in the direction of reduced free energy, not to gain higher free energy. In the second paragraph, the authors get into a mode of extreme generalization, for instance suggesting (line 54) that liquid-liquid phase separation (LLPS) would require the presence of macromolecules and water; neither is required, as LLPS is very general, with many examples in mixtures of organic small molecules, without any water.

Response: We appreciate the Reviewer for pointing out the mistakes and lack of rigor in our manuscript. We have thoroughly reviewed and corrected these errors to ensure clarity and prevent any potential misinterpretation of the fundamental concepts presented.

The relevant discussion in the revised manuscript has been rephrased as:

In abstract: *“In the current work, we develop a series of heterogeneous colloidal suspensions that exhibit both liquid-liquid phase separation (LLPS) of semiflexible binary polymers and liquid crystal phase separation (LCPS) of rigid, rod-like nanocellulose particles.”*

On page 3: *“To minimize free energy, certain kind of colloids can demix into two coexisting phases to reach equilibrium.”*

On page 3: *“By contrast, the underlying driving force of LLPS is the trade-off between enthalpy and entropy toward energy minimization, based on the interactions between different components.”*

6. Fig. 1, quite central to the discussion throughout the paper, is full of ambiguity and/or errors. Panel (a) is said to show a “volume-composition diagram” but there is no volume axis in this diagram; it is a 1D diagram, the only variable being composition, increasing from left to right.

Response: We thank the Reviewer for pointing out our lack of rigor. We have revised the phase diagram illustration (see **Figure 12** and Figure 1a in the revised manuscript).

Figure 12. The volume-composition diagram illustrates two vertical lines at Onsager volume fractions ϕ_I and ϕ_{LC} , corresponding to anisotropic colloidal particles with either one-phase or two-phase systems.

7. The diagram in panel (b) is incomprehensible: What is the black curve? What are the coils drawn in the region labelled “Two phases”? Are there no polymer coils in the single-phase system? If the black curve is a binodal coexistence curve, there are no states at all in the middle, between the two flanks below the plait point. Where is the temperature or pressure that are mentioned in the text (line 55) that refers to this diagram? Where is the temperature or pressure that are mentioned in the text (line 55) that refers to this diagram? What are the compositions of the two coexisting phases? What happens above the peak of the black curve; are the authors suggesting that there is only one phase at high enough concentration? Why is then the bottom, low-concentration, region labelled as “One phase”?

Response: We sincerely thank the Reviewer for all the questions regarding to Figure 1b. We agree that our description was quite limited, and the figure needed revision. We carefully addressed each of the raised questions and concerns. We have generated a new schematic illustration of the phase diagram to explain how temperature and composition affect LLPS transitions (see **Figure 13**).

Figure 13. A phase diagram describes the phase behavior of macromolecular solutions as a function of concentration, influenced by various modulatory factors such as temperature and pH etc. The coexistence line (black curve) separates the one-phase region from the two-phase region on the phase diagram. At the critical point, the composition of the two liquid phases becomes identical, and the density difference between the phases approaches zero. Phase separation starting from a homogeneous

state (red dot) can be induced by increasing the concentration ($\Delta C > 0$) or decreasing the temperature ($\Delta T < 0$). At concentrations below saturation concentration (C_{sat}), the system is in the one phase region (1). Within the two-phase region, a polymer solution forms dense droplets with enriched solutes and a depleted dilute phase (2). As concentration increases, droplets grow (3) until the dense phase volume surpasses the diluted, causing diluted droplets to form within the dense phase (4). Past this inversion, increasing concentration shrinks the diluted droplets until only a dense one-phase solution remains (5).

8. In the caption to panel (c), the authors write that the two lines are coexistence lines, but that cannot be true because such lines indicate which two phases coexist; there is no such indication here. Coexistence lines are lens- or bell-shaped (similar to the black curve in panel (b)). Moreover, what is the physical origin of these lines in panel (c)? Or are they simply hand-drawn? The caption also refers to "the binodal curve" without such a curve ever having been introduced; where is it? A binodal curve is bell-shaped with a plait point at the top or at the bottom. I see no such curve in panel (c).

Response: We thank the Reviewer for the questions and apologize for the confusion. Figure 1c is a generic phase diagram for an aqueous solution of PEG and dextran, with the X and Y axes representing the concentration of dextran and PEG, respectively. The black and red dashed curves are intended to represent the coexistence line separating the one-phase and two-phase regions at a certain temperature. This type of phase diagram is commonly used in many previous studies relevant to LLPS transitions (*Accounts of Chemical Research*, **2012**, *45*, 2114–2124. *Chemical Society Reviews*, **2020**, *49* 114–142.). In our study, we aimed to emphasize that temperature changes can alter the critical polymer concentrations for LLPS transitions. The red dashed line was drawn to represent the new coexistence curve at an altered temperature. Consequently, the PEG-dextran solutions with the composition at the intermediate region between the two coexistence lines stay in two phases due to its composition is above the coexistence curve (black curve), but remix into one phase at a specific temperature due to the composition is below the second coexistence curve (red dashed curve).

We have redrawn the coexistence curves with lens-shape (see **Figure 14** and Figure 1c in the revised manuscript) and enhanced the figure caption in the revised manuscript.

Figure 14. Phase diagram of PEG-dextran LLPS system: the binodal curve separates the phase diagram into a one-phase region that below the curve and a two-phase region that above the curve. The black curve is the coexistence line at the given temperature, whereas the red dashed curve represents a new coexistence line at an altered temperature. When the composition of PEG-dextran solution is located at the intermediate region between the two coexistence lines, it will stay in two-phase state at a given temperature and remix into one phase with temperature variation.

9. The confusion continues on page 5, for instance with the idea (line 85) that cooling would “shift” a binodal phase diagram toward lower concentration of the dissolved polymers PEG and Dextran; what they are doing is shifting their system into a different part of the phase diagram. The same problem reappears on page 8, line 145 in the text “the phase diagram of CNC-PEG-dextran mixtures had shifted in the favor of a two-phase over a one-phase system”. The phase diagram does not shift; the authors are shifting their sample composition such that it ends up in a different part of the four-component phase diagram.

Response: We thank the Reviewer for the academic accuracy and strictness and apologizing for the confusion caused by our previous wording. Our description is not correct, and we fully agree that that the phase diagram does not shift, but rather the sample composition positions the sample in a different region of the phase diagram. We have revised the manuscript to clarify this point and accurately convey our findings. The relevant discussion in the revised manuscript has been rephrased as:

“Intriguingly, this LLPS is temperature sensitive: it can be triggered by cooling when shifting the PEG-dextran composition from one-phase into the region that favors phase separation in the phase diagram.”

“However, the four-component CNC-PEG-dextran aqueous mixtures (x wt%-3.75 wt%-4.25 wt%, $x=1-3$) demonstrated a two-phase stacking without liquid crystalline ordering (Figure S10), implying that the addition of CNC had shifted the PEG-dextran composition in favor of a two-phase over the one-phase state in the phase diagram.”

10. It is hardly surprising that you see biphasic regimes as a result of this. To understand what is going on, in particular in the following experiments where the CNC fraction increases, it would be important to know what the phase diagram looks like and what the compositions of the different phases are. This is actually a key problem with this paper: they never bother to even sketch a phase diagram or to clearly define how each phase is composed. Without some kind of diagrams explaining what is going on as compositions change in their 4-component system (that’s another mistake on page 5, line 76; they write that it is a 3-component system, apparently forgetting the solvent, which is extremely important for the behavior), every discussion of different phases remains extremely vague. Admittedly, representing 4-component phase diagrams is not easy, and obviously requires defining a set of representative 2- or 3-component phase diagrams, where the other one or two components are kept constant (as well as temperature, which adds a fifth variable to the study, since they also vary temperature). But it was the authors’ choice to go for such a complex system; they need to break it down into pieces that can be represented, conducting their experiments accordingly and referring to each 2- or 3-dimensional diagram in analyzing the outcomes.

Response: We thank the Reviewer whose suggestions helped us to re-design the phase diagrams for the complex system at hand. We have amended the manuscript and defined the system as a four-component throughout the text, to reflect the role of water. Within the four-component system, CNCs undergo a LCPS transition, while PEG and dextran partake in LLPS. The multiphase separation behavior of CNC-PEG-dextran aqueous mixture is a result of the interplay between the two phase separation processes, affected by variations in temperature.

Based on the Reviewer’s suggestion, we have conducted extensive measurements to find the critical compositions for CNC-PEG-dextran suspension that undergoes multiphase separation behavior at either 21 or 50 °C. We found that the fraction of CNC significantly affects the critical PEG-dextran compositions for LLPS and governs the multiphase transitions across different phase regions. In order to better present these results, we have mapped the three-dimensional phase diagrams, which clarify the critical composition of CNC-PEG-dextran for multiphase separation and identify the physical origin of each phase behavior (see **Figure 15** and Figure 3 of the revised manuscript). The obtained phase diagram not only create a roadmap to demonstrate the multiphase separation behavior of four-component CNC-PEG-dextran aqueous mixture at different concentrations, but also reveals the physical origin of two-phase, three-phase and four-phase behaviors in different categories.

Figure 15. Three-dimensional phase diagram of the CNC-PEG-dextran aqueous mixture that highlight the interplay between LCPS of CNCs and LLPS of polymers at 21 °C (a) and 50 °C (b), respectively. Among which, the LLPS process is inhibited at the left side of the CNC-PEG-dextran coexistence plane (gray dots) whereas the right side area is active for LLPS. With the contribution of LCPS, the resulting multiphase regions are modulated by the critical CNC concentrations (colored horizontal planes). (c) When the CNC concentration exceeds 4.5 wt%, the multiphase transition area can be defined as two-phase region (only LCPS, points 1 and 4), inverted phase region (where a cholesteric phase is on top of an isotropic phase, points 2 and 5), and a four-phase region (points 3 and 6) due to changes of the polymer composition.

The relevant discussion in the revised manuscript has been rephrased as:

“To better understand the multiphase separation, we clarify the interplay between LLPS and LCPS by mapping the three-dimensional phase diagrams for the four-component CNC-PEG-dextran aqueous mixtures at 21 and 50 °C, respectively (Figure 3). When the CNC concentration is below 3.25 wt% and equilibrate under 21 °C, the phase diagram of CNC-PEG-dextran aqueous mixture can be divided into one-phase region and two-phase region by the coexistence plane (gray dots). The obtained mixtures remain homogeneous with the compositions located at the left side of the CNC-PEG-dextran coexistence plane and phase separated due to the contribution of LLPS process at a composition on the right side of the coexistence plane (Figure 3a). The phase separation behavior becomes more pronounced by increasing the CNC concentration, which is ascribed to the occurrence of LCPS. Given the multiphase separation process, the obtained CNC-PEG-dextran aqueous mixture display either a two-phase or three-phase behavior. It should be noted that the origin of the two-phase behavior can be realized by the contribution of LCPS or coupled LLPS-LCPS. Further increasing CNC concentration

to 5.5 wt% leads to the emergence of an additional cholesteric phase within the PEG-rich region, thereby transforming the hybrid system from a two-phase to a three-phase state.

Upon heating, the interactions of CNC-polymers and PEG-dextran changed, leading to temperature-sensitive multiphase separation with two-, three- and four-phase coexistence behaviors of different physical origins (Figure 3b). To explore the multiphase separation under 50 °C, the three-dimensional phase diagram can be partitioned by the CNC-PEG-dextran coexistence plane and the critical CNC concentration planes into several regions. With the addition of polymers, the LCPS of CNC is profoundly suppressed; namely, the CNC concentration threshold for LCPS shifts from 3.25 wt% (21 °C) to 4.50 wt% (50 °C) in the polymer miscible region. However, in the region where LLPS occurs, the critical concentration for LCPS of CNC remains at 3.25 wt%. Above this critical CNC concentration, the four-component CNC-PEG-dextran aqueous mixtures demonstrate three-phase behavior with LLPS and LCPS process, including an upper PEG-rich isotropic phase, a middle dextran-rich isotropic phase, and a bottom dextran-rich cholesteric phase. Intriguingly, when the CNC concentration reaches 4.50 wt%, the resulting mixtures exhibit more complex phase behavior with the emergence of varying inverted cholesteric-isotropic states. Figure 3c highlight the main critical coexistence plane (gray dots) and secondary phase plane (magenta square) with a transition from two-phase, three-phase to four-phase states, where the intermediate region between the two phase boundaries displays an unique inverted phase behavior. Within this region, the volume fraction of upper cholesteric PEG-rich phase increases with the increasing of CNC concentration, and finally the multiphase system shifts from three-phase to two-phase stacking (from point 5 to 2). Further changes in the polymer concentration, below the main critical coexistence plane or beyond the secondary phase plane, leads to two-phase (point 1 and 4) and four-phase behaviors (point 3 and 6), respectively.”

11. An example of a situation where an understanding of the compositions of the separating phases is critical to a correct interpretation is found on page 7, line 127, where the authors conclude that liquid crystal phase separation (LCPS) could be triggered “not only by the concentration, but also modified by adding semiflexible nonadsorbing polymers”. However, as is clear from the above-mentioned works by Dogic and co-workers, the impact of depletion attraction on a colloidal system developing LC phases is to nucleate phases with much higher local concentration of the particles than in the global system. In other words, since the authors do not take the different concentration of CNC in the separating LC phase into account, they incorrectly conclude that concentration would not play a role in the liquid crystal phase separation.

Response: The concentration of CNCs indeed plays a significant role in the LCPS process. We never drew the conclusion that concentration would not influence the LCPS. The relevant discussion has already shown in Figure 2a in our previous manuscript (see **Figure 16** below), indicating the concentration dependent LCPS of pure CNC suspension. When the CNCs concentration reaches 4 wt%, we observed a bottom anisotropic liquid crystal phase and an upper isotropic phase that due to the LCPS of CNC. Moreover, as a comparison, we also investigated the influence of different CNC concentration on the phase behavior of CNC-polymer aqueous mixtures by fixing the polymer composition, while the CNC concentration was tuned from 0 to 6 wt%.

Figure 16. Photograph of aqueous CNC suspensions to highlight the concentration dependent LCPS (samples were placed between crossed polarizers).

In order to address the Reviewer's concern, the relevant discussion in the revised manuscript has been rephrased as:

“Therefore, we concluded that the LCPS of CNCs could be triggered by increasing concentration and further modified by adding semiflexible nonadsorbing polymers, giving rise to a concentration-driven, temperature-sensitive phase behavior.”

12. The idea to contrast LLPS against LCPS is not necessarily appropriate. On page 8, line 147, they write that “both LLPS of polymers and LCPS of CNCs occurred in the hybrid system”, but it is not correct to attribute LLPS to polymers and LCPS to CNCs. All components influence the behavior of the system. The authors demonstrated just before that the addition of a small amount of CNCs can induce LLPS without any liquid crystal formation. Likewise, the presence of polymers will influence the liquid crystal formation and related phase separation. With two types of polymers dissolved into the solvent of a polydisperse nanoparticle suspension, they have created a very complex system, and they have to consider the impact of every component on every phenomenon.

Response: We fully agree with the Reviewer's comment that every component in the CNC-PEG-dextran aqueous mixtures have strong influence on the final multiphase behavior and there is an interplay between LLPS of polymers and LCPS of CNCs. As shown in the three-dimensional phase diagrams (see **Figure 15**), the addition of CNCs indeed alters the critical polymer concentrations necessary for LLPS, while the presence of polymer compositions also profoundly impact the LCPS process, indicating the interdependency between the two transition processes.

However, when we refer to the LLPS of polymers or the LCPS of CNCs, we aimed to emphasize the main physical contribution of LLPS is due to PEG and dextran in the multiphase system, whereas the LCPS process comes from the formation of CNC cholesteric assembly. This kind of expression can provide us a more straightforward analysis on the multiphase separation behavior that resulting from the combined influences of all components.

13. On page 18, line 318, the sentence “As the multiphase separation continues, LLPS reach equilibrium and LCPS begin” is problematic. This would suggest that the initial LLPS gives rise only to isotropic phases, and only when the “LCPS begin”, an anisotropic liquid crystalline phase would be seen. However, the long paragraph preceding this sentence, supposedly describing LLPS, refers to Fig. S18–S21, which all clearly reveal the presence of liquid crystal phases already at this stage. The DIC images in Fig. S18 and S21 have many areas with fingerprint lines typical of cholesteric phase formation, and the POM images in Fig. S19-20 are very colorful, clearly demonstrating that the phases are birefringent. The idea that first LLPS would complete and reach equilibrium, and only after this LCPS would take place is thus contradicted by the experimental evidence.

Response: We thank the Reviewer for pointing out this mistake. In fact, both LLPS and LCPS simultaneously proceed once the samples are allowed get equilibrium. Both transition processes share the nucleation-growth pathways, but the dynamics of the two separation processes are different. Our results indicate that the LLPS process has faster phase separation kinetics than the LCPS one (see **Figure 17**). During the initial stage of phase separation (0-1.5h), LLPS was leading to the formation of PEG-dextran interface, the LCPS process concurrently undergoes nucleation of CNCs into anisotropic tactoids. At this stage, no macroscopic isotropic-cholesteric interface was generated. The formation of isotropic-cholesteric interface happened *ca.* 2 h later. Therefore, LLPS reaches equilibrium prior to the observable onset of LCPS at a macroscopic level.

The relevant discussion in the revised manuscript has been rephrased as:

“As the multiphase separation continues, LLPS reaches equilibrium while LCPS is still evolving.”

Figure 17. Temperature-dependent multiphase separation kinetics measured under 21 °C (a) and 50 °C (b), displaying individual LLPS and LCPS processes. Phase separation velocity (dV/dt) of individual LLPS and LCPS transition calculated from kinetic data at 21 °C (c) and 50 °C (d).

14. Considering the importance of PEG and Dextran in this study, I am surprised that the authors never discuss why a ternary mixture of PEG and Dextran in water will tend to separate at certain temperatures but stay homogeneous at others, and why their interaction with CNCs is different. I imagine that the sugar ring structures in Dextran give it a more favorable interaction with CNC than PEG, but the authors never discuss this.

Response: We thank the Reviewer for the good comment. The reviewer raises very good points.

1. The thermos-responsive phase behavior of PEG-dextran is attributed to the interplay between enthalpy and entropy. Previous study demonstrate that the driving force can be attributed to the entropic gain associated with the release of water molecules from the polymer chains upon heating (*J. Chromatogr. B: Biomed. Sci. Appl.* 1998, **711**, 3–17).
2. As depletants, PEG and dextran displayed varying affinity for CNCs due to the molecular differences between PEG and dextran. Compared with PEG, dextran molecule is composed of branched glucose units that similar to the molecular structure of CNC (a linear chain glucose units), appear to show more favorable interactions between CNC-dextran than CNC-PEG (Kittle, Joshua Daniel. *Characterization of cellulose and chitin thin films and their interactions with bio-based polymers*. Diss. Virginia Tech, **2012**). This hypothesis can be further confirmed by POM analyzing the CNC-PEG and CNC-dextran cholesteric phase (see **Figure 11** and Figure S4 in the revised manuscript), as discussed above.

Based on the reviewer’s comment, the relevant discussion regarding to these issues have been rephased as:

“The driving force behind this phenomenon can be attributed to the entropic gain associated with the release of water molecules from the polymer chains.”

“In such aqueous CNC-polymer mixtures, CNCs are co-assembled with polymer coils into cholesteric organization and creating a region that is depleted of polymers around each nanorods (Figure 2b and Figure S3). Owing to the molecular differences between PEG and dextran, the microscopic texture of CNC-PEG and CNC-dextran liquid crystal phase are slightly different under depletion effect (Figure S4). This maybe because of dextran (composed by branched glucose units) share similar molecular

structure with CNCs (composed by linear chain glucose units), leading to more favorable steric and intermolecular interactions.”

15. Likewise, I cannot find the evidence for their statements of a certain phase being PEG-rich and another being Dextran-rich; they just write, for instance on line 149, that the bottom phase is Dextran-rich and the upper is PEG-rich, but where is the experimental evidence? This question recurred throughout the manuscript while I was reading.

Response: The polymer composition of each phase in the equilibrated CNC-PEG-dextran aqueous mixtures is determined through size-exclusion chromatography (SEC), as shown in **Figure 18** (see Figure S7 in the revised manuscript). The accurate polymer composition of each phase is shown in **Table 1**, revealing the presence of two distinct subcompartments separated by the polymer-polymer interface. Detailed calculations for determining the polymer composition can be found in the Supporting Information. These results were mentioned in the original manuscript on page 9: *After equilibrium, we determined the relative composition of each phase by size-exclusion chromatography and turbidity measurements (Figure S5-S7 and Table S2), indicating that the upper PEG-rich and bottom dextran-rich subsystems possess unequal affinity to CNCs.*

Figure 18. The size-exclusion chromatography for pure CNC, PEG and dextran with the injected concentration of 1 mg/mL, respectively. Dextran is eluted earlier with a peak at 36.95 minutes, while the peak for PEG appears later at 44.25 minutes (a). The dependence of the RI peak area A_{RI} as a function of the polymer concentration (b). The size-exclusion chromatography of each phase in the CNC-PEG-dextran aqueous mixtures with different initial compositions: (c), (d), (f) and (h) were equilibrium under 21 °C, while (e), (g) and (i) were equilibrium under 50 °C, respectively.

Table 1 Relative composition of each phase in the equilibrated CNC-PEG-dextran aqueous mixtures based on size-exclusion chromatography data.

	Initial Composition (wt%)			Density (g/cm ³)	Concentration (mg/mL)		
	PEG	Dextran	CNC		PEG	Dextran	
21 °C	3.75	4.25	0	PEG-rich phase	1.025	40.82	31.36
				Dextran-rich phase	1.063	21.23	72.99
	3.75	3.5	6	PEG-rich isotropic phase	1.025	42.23	17.85
				PEG-rich cholesteric phase	1.175	50.00	13.57
				Dextran-rich cholesteric phase	1.242	17.13	81.20
				Dextran-rich isotropic phase	1.032	44.90	8.28
3.75	4.25	6	PEG-rich cholesteric phase	1.070	50.38	11.11	
			Dextran-rich cholesteric phase	1.238	14.22	96.68	
50 °C	3.75	3.5	6	PEG-rich cholesteric phase	1.193	43.23	26.83
				Dextran-rich isotropic phase	1.234	21.98	66.88
	3.75	4.25	6	PEG-rich isotropic phase	1.138	56.43	14.58
				PEG-rich cholesteric phase	1.208	54.36	15.75
				Dextran-rich isotropic phase	1.235	25.18	70.46
				Dextran-rich cholesteric phase	1.270	21.97	75.67

16. On page 11, lines 204–206, the authors write that a fingerprint texture of one phase is “continuously invaded” from another cholesteric phase, without presenting any picture evidence. It is unclear what is meant by this.

Response: We apologize for our poor description. We had taken polarized optical microscopy (POM) images of the equilibrated three-phase stacking at 21 °C, specifically focusing on the microstructure of the polymer-polymer interfaces, as presented in Figure 3a and b in the original manuscript. We observed that the fingerprint texture of the cholesteric phase in the PEG-rich phase continuously propagates into the dextran-rich phase (see **Figure 19**).

To enhance the clarity and provide better visual evidence, we have included **Figure 19** as Figure S17 in the revised Supporting Information.

Figure 19. High magnified POM images that focus on the PEG-dextran interface of the equilibrated four-component CNC-PEG-dextran aqueous mixture at 21 °C, showing continuous fingerprint texture propagates from the PEG-rich phase into the dextran-rich phase. The initial composition of the mixtures is 6 wt%-3.75 wt%-4.25 wt% for (a) and 6 wt%-3.75 wt%-3.5 wt% for (b).

17. They then go on by describing the interface as being an “immiscible PEG-dextran interface”. What is meant by “immiscible” here? An interface can neither be miscible nor immiscible. Do they mean that the two sides of the interface contain only PEG and Dextran, respectively? That will hardly be the case, as phase separation never leads to separation of a pure component, unless that component is crystallizing out as a solid crystal, which is not the case here.

Response: We thank the Reviewer for the opportunity to clarify this point. LLPS of PEG-dextran solution can create two “compartments” with an upper PEG-rich (with a small amount of dextran) phase and a bottom dextran-rich (with a small amount of PEG) phase. From the chemical point of view, both phases have the same components composition, but they do not mix. From the physical point of view, both PEG-dextran interface and isotropic-cholesteric interface of pure CNC suspension are related to water-water interfaces with ultralow interfacial surface tension, however the latter are miscible to each other. Therefore, we refer to the PEG-dextran as “immiscible phases”.

In order to address the Reviewer’s comment, the relevant discussion has been corrected as:

“When viewed under higher magnifications, we observed two sharp interfaces at the isotropic-cholesteric and cholesteric-cholesteric domains in which the fingerprint texture in the latter region was continuously invaded from the middle cholesteric phase to the bottom phase, implying the chiral assembly of CNCs across the PEG-dextran phase boundary where the two phases are immiscible to each other.”

18. The confusion regarding the separation phase compositions gets rather extreme at some points, like line 297-298 on page 16, where the authors write that LLPS-induced nuclei would be “polymer droplets”; of course they are not pure droplets of polymer, but probably a liquid solution enriched in one or both polymers.

Response: We thank the Reviewer for pointing out our lack of rigor. In the revised manuscript, the relevant discussion has been rephrased as: *“Once LLPS occurs, the system first becomes cloudy due to the nucleation of polymer-rich water droplets and shows no significant decreasing of volume fraction in the frozen stage.”*

19. Also the descriptions of data are incomplete. For instance, to the many images of samples in which phase separation occurs, the authors write in the caption only that they are “photographs”. How were they taken? Are the samples between crossed polarizers? If not, how do the authors distinguish between liquid crystalline and isotropic phases?

Response: We thank the Reviewer for bring this issue to our attention. The liquid crystallinity of these samples was verified by either POM analysis or taken photos under crossed polarizers. For example, the images of pure CNC suspension with different concentration in Figure 2a were taken under crossed polarizers. In Figure 2d and e, the photographs of the equilibrated samples were taken without crossed polarizers, but we verified the liquid crystallinity of these samples through using POM analysis.

Referee 3:

1. This paper described a controllable phase transition coupled with temperature-dependent liquid-liquid phase separation (LLPS) and concentration-dependent liquid crystal phase separation (LCPS). The observation of the assembly process of cellulose nanocrystals (CNCs) in the compartmentalized polymer regions appeared convincing. The analysis of the influencing factors of the phase behavior was done with great care. The conclusions were drawn based on appropriate controls and sufficiently strong evidence. The list of references was comprehensive and well placed. The paper was, in most parts, pleasant to read and the reasoning was easy to follow. It presented an advancement in the understanding of the underlying science of the similar systems (their early publication in ACS Nano).

Response: We appreciate the summary and positive feedback on our work of our manuscript.

2. Having said that, the work would be more appealing if a proof-of-concept illustration can be presented to showcase the applications of their studies.

Response: We thank the Reviewer for this wonderful suggestion. We have expanded current system to mimic some nature phenomenon observed in biology. However, these investigations are still in progress, and we think would extend the paper, already quite complex, less focused.

3. Wordings in Figure 3 were too small to read.

Response: We thank the Reviewer for pointing out this issue. The original Figure 3 has been changed to Figure 4 in the revised manuscript. We have made the necessary changes to the front size in Figure 4 to improve its readability. The font size in Figure 4 has been increased to improve the readability.

4. Scale bars in Figures 3-5 were partially labelled.

Response: We thank the Reviewer for the attention to details. Based on the Reviewer's concern, we have removed the length labels from the scale bars and attached a note in the figure caption to specify the scales.

5. 'SAXS' in Figure S16 was miss-spelt.

Response: We are sorry for our typo. The spelling error has been corrected in the revised Supporting Information.

6. In line 58 of SI, 'pith' was miss-spelt.

Response: We are sorry for our typo and really appreciate the reviewer's careful reading. We have corrected this spelling error in the revised Supporting Information.

7. I recommend acceptance for publication after satisfactory addressing of the above.

Responses: Owing to the several additional experiments as described above for addressing the comments and criticisms raised by the Reviewers, we believe that our manuscript is now considerably polished up and suitable for Nature Communications.

REVIEWERS' COMMENTS

Reviewer #1 (Remarks to the Author):

[Note from the Editor: Reviewer #1 made comments to the editor only. Reviewer #1 was asked to look also over the response given to reviewer #2 and thinks that despite the complex and challenging system, the authors addressed the technical concerns of the reviewers]

Reviewer #3 (Remarks to the Author):

The authors have addressed the comments, leading to improved quality of the manuscript. Therefore, I recommend acceptance of the manuscript for publication.